# Periodic Activation Functions Induce Stationarity

**Lassi Meronen**
Aalto University / Saab Finland Oy
Espoo, Finland
lassi.meronen@aalto.fi

**Martin Trapp**
Aalto University
Espoo, Finland
martin.trapp@aalto.fi

**Arno Solin**
Aalto University
Espoo, Finland
arno.solin@aalto.fi

## Abstract

Neural network models are known to reinforce hidden data biases, making them unreliable and difficult to interpret. We seek to build models that 'know what they do not know' by introducing inductive biases in the function space. We show that *periodic* activation functions in Bayesian neural networks establish a connection between the prior on the network weights and translation-invariant, stationary Gaussian process priors. Furthermore, we show that this link goes beyond sinusoidal (Fourier) activations by also covering triangular wave and periodic ReLU activation functions. In a series of experiments, we show that periodic activation functions obtain comparable performance for in-domain data and capture sensitivity to perturbed inputs in deep neural networks for out-of-domain detection.

## 1   Introduction

Deep feedforward neural networks [46, 16] are an integral part of contemporary artificial intelligence and machine learning systems for visual and auditory perception, medicine, and general data analysis and decision making. However, when these methods have been adopted into real-world use, concerns related to robustness (with respect to data that has not been seen during training), fairness (hidden biases in data being reinforced by the model), and interpretability (why the model acts as it does) have taken a central role. The knowledge gathered by contemporary neural networks has even been characterised as never truly reliable [31]. These issues relate to the sensitivity of the trained model to perturbed inputs being fed through it—or the lack thereof.

This motivates *Bayesian deep learning*, where the interests are two-fold: encoding prior knowledge into models and performing probabilistic inference under the specified model. We focus on the former. Probabilistic approaches to specifying assumptions about the function space of deep neural networks have gained increasing attention in the machine learning community, comprising, among others, work analysing their posterior distribution [57, 1], discussing pathologies arising in uncertainty quantification [9], and calls for better Bayesian priors (*e.g.*, [42, 50, 37, 10]).

In this paper, we focus on stationary models, which act as a proxy for capturing *sensitivity*. Stationarity indicates translation-invariance, meaning that the joint probability distribution does not change when the inputs are shifted. This seemingly naive assumption has strong consequences in the sense that it induces *conservative* behaviour across the input domain, both in-distribution and outside the observed data. The resulting model is mean-reverting outside the training data (reversion to the prior), directly leading to higher uncertainty for out-of-distribution (OOD) samples (see Fig. 1 for examples). These features (together with some direct computational benefits) have made stationary models/priors the standard approach in kernel methods [4, 20], spatial statistics [5], and Gaussian process (GP) models [44], where the kernel is often chosen to induce stationarity.

Neural networks are parametric models, which typically fall into the class of non-stationary models. Non-stationarity increases *flexibility* and is often a sought-after property—especially if the interest is solely in optimizing for accuracy on in-domain test data. In fact, all standard neural network

35th Conference on Neural Information Processing Systems (NeurIPS 2021).

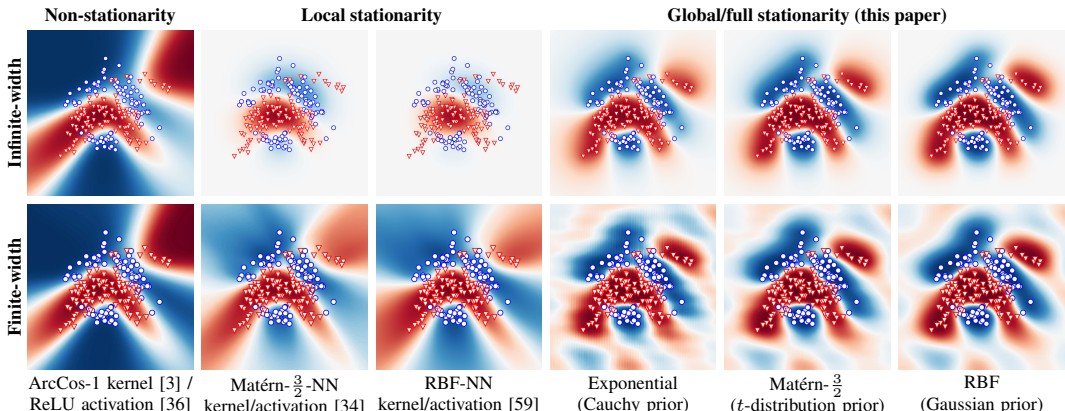

Figure 1: Posterior predictive densities of single hidden layer Bayesian neural networks (BNNs) with 30 hidden units and their infinite-width corresponding GPs on the banana classification task. Different activation functions induce different prior assumptions. Estimates obtained through HMC sampling [14] for 10k iterations.

activation functions (ReLU, sigmoid, *etc.*) induce non-stationarity (see [3, 59]). However, non-stationary models can easily result in over-confidence outside the observed data [18], [55] or spurious relationships between input variables (as illustrated in Fig. 1, where darker shades show higher confidence).

Stationarity in neural networks has been tackled before under the restriction of local stationarity, *i.e.*, translation-invariance is only *local*, induced by a Gaussian envelope (as realized by [59] for the RBF kernel/activation). Recently, Meronen et al. [34] expanded this approach and derived activation functions corresponding to the widely used Matérn class of kernels [32, 44]. We go one step further and derive activation functions that induce *global* stationarity. To do so, we leverage theory from harmonic analysis [58] of periodic functions, which helps expand the effective support over the entire input domain. We also realize direct links to previous works leveraging harmonic functions in neural networks [13, 2, 61, 47, 56], and Fourier features in other model families [43, 51].

The contributions of this paper are: *(i)* We show that periodic activation functions establish a direct correspondence between the prior on the network weights and the spectral density of the covariance function of the limiting stationary Gaussian process (GP) of single hidden layer Bayesian neural networks (BNNs). *(ii)* We leverage this correspondence and show that placing a Student-$t$ prior on the weights of the hidden layer corresponds to a prior on the function space with Matérn covariance. *(iii)* Finally, we show in a range of experiments that periodic activation functions obtain comparable performance for in-domain data, do not result in overconfident predictions, and enable robust out-of-domain detection.

## 1.1 Related Work

We build upon prior work on exploring the covariance function induced by different activation functions, starting with the seminal work by Williams [59], who discussed a sigmoidal (ERF) and a Gaussian (RBF) activation function, resulting in a locally stationary kernel modulated by a Gaussian envelope. Cho and Saul [3] introduced non-stationary GP kernels corresponding to the ReLU and step activations, and [54] later extended the approach to the leaky ReLU and analysed alternative weight prior specifications. More recently, [34] derived activation functions corresponding to kernels from the Matérn family, which are locally stationary modulated by Gaussian envelope. In addition to the work connecting neural networks to GPs at initialisation, [21] created a connection between the neural network training and kernel methods by introducing the Neural Tangent Kernel (NTK).

At the same time, uninformative priors have been widely criticised in Bayesian deep learning [10], and alternative strategies have been proposed to incorporate prior knowledge. Pearce et al. [42] proposed compositions of BNNs to mimic compositional kernels in their limiting case. Sun et al. [50] proposed a variational objective for BNNs acting on the function space rather than the network parameters. Nalisnick et al. [37] proposed tractable priors on the functional complexity, obtained through the change of variables formula applied to the KL-divergence from a reference model/prior.

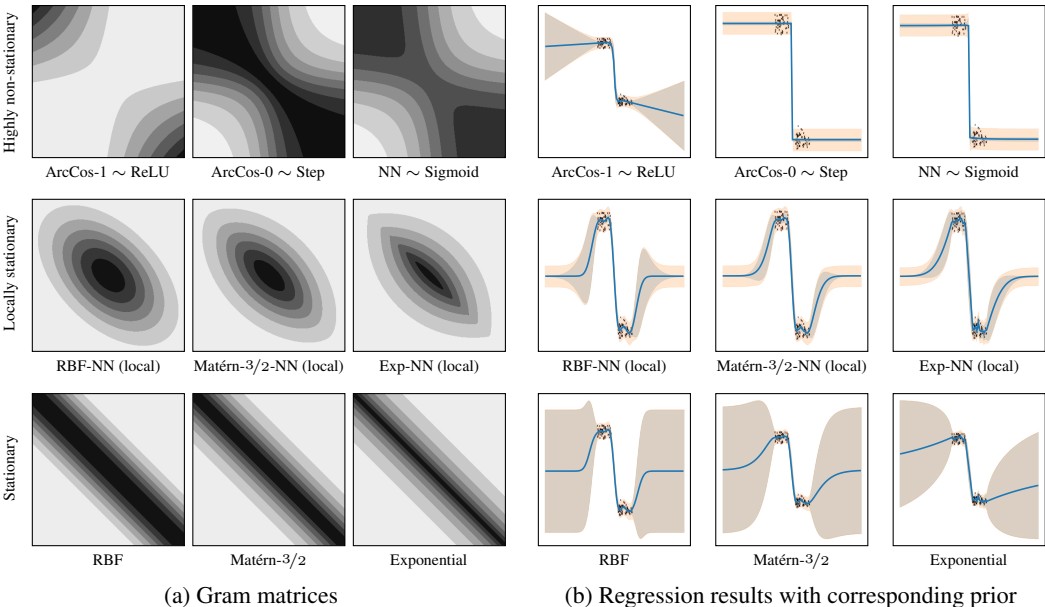

(a) Gram matrices                  (b) Regression results with corresponding prior

Figure 2: **Left:** Gram/covariance matrices (for $\kappa(x, x')$) reflecting different prior beliefs on how infinite-width NN model functions ought to behave outside the observed data. **Right:** 1D regression results corresponding to the model induced by the priors, showing the posterior ▇ and predictive ▇ 95% intervals. See Fig. 8 in appendix for the corresponding finite-width results.

Furthermore, Morales-Alvarez et al. [35] proposed to encode prior assumptions on the activation level rather than the model parameters.

Recent works have shown the potential of periodic activation functions in neural networks, including approaches to learn implicit neural representations [47], methods for recognition of handwritten digits [61], and out-of-distribution detection [30]. Earlier work used periodic activation functions to resemble the Fourier series [13, 56] and showed that such models are universal function approximators [2]. In broader terms, [27] recently showed that the use of the Fourier transform, replacing self-attention, in Transformers can drastically improve the runtime. Inspired by the Fourier duality of covariance and spectral density, [43] proposed random Fourier features to approximate a GP with an RBF kernel using Monte Carlo integration. [51] later analysed the quality of the approximation, and various works have explored (random or deterministic) Fourier features, *e.g.*, in the context of kernel methods (*e.g.*, [24, 19, 48]). [53] investigated the use of Fourier features in multilayer perceptrons (MLPs) to help represent high-frequency functions.

To the best of our knowledge, none of the previous works has shown that periodic activation functions in BNNs establish a direct correspondence between the prior on the network weights and the density of the spectral density decomposition, thus, inducing global stationarity and enabling principled prior choices for BNNs.

## 2 Encoding Inductive Biases into Infinitely-wide Networks

We are concerned with the notions of *similarity* and *bias* in the input space when performing supervised learning. For this, we consider modelling assumptions present in random (untrained) models that induce the *a priori* (before observing data) characteristics in the model. We argue that these assumptions, when combined with data, either reinforce or suppress data biases depending on the prior assumptions. For analysis, we consider the mean function, $\mu(\cdot)$, and covariance (kernel) function, $\kappa(\cdot, \cdot)$, induced by model functions $f(\mathbf{x})$, representing the input–output mapping of a single-layer: $\mu(\mathbf{x}) \coloneqq \mathrm{E}[f(\mathbf{x})]$ and $\kappa(\mathbf{x}, \mathbf{x}') \coloneqq \mathrm{E}[(f(\mathbf{x}) - \mu(\mathbf{x}))(f(\mathbf{x}') - \mu(\mathbf{x}'))^*]$, where $\mathbf{x}, \mathbf{x}' \in \mathbb{R}^d$ are two inputs and the expectations are taken over model functions. In probabilistic machine learning, rather than inferring the covariance structure from the expectations over model functions, one typically directly encodes assumptions by choosing a suitable parametric kernel family. However, both approaches, building a model by choosing a form for $f$ or choosing a kernel family, are equivalent.

Specification of prior knowledge through the covariance function and doing inference under this model, is at the core of Gaussian process (GP) models [44]. These models admit the form of a prior $f(\mathbf{x}) \sim \mathcal{GP}(\mu(\mathbf{x}), \kappa(\mathbf{x}, \mathbf{x}'))$ and a likelihood (observation) model $\mathbf{y} \mid \mathbf{f} \sim \prod_{i=1}^{n} p(y_i \mid f(\mathbf{x}_i))$, where the data $\mathcal{D} = \{(\mathbf{x}_i, y_i)\}_{i=1}^{n}$ are input–output pairs and $\mathbf{x}_i \in \mathbb{R}^d$. This non-parametric machine learning paradigm covers regression and classification tasks, under which GPs are known to offer a convenient machinery for learning and inference, while offering meaningful uncertainty estimates.

Neal [38] showed that an untrained single-layer network converges to a GP in the limit of infinite width. This link is well-understood and generalizes to deep models [7], [26]. By placing an i.i.d. Gaussian prior on the weights and biases of a network, the distribution on the output of an untrained network converges to a Gaussian distribution in the limit of infinite width. By application of the multivariate Central Limit Theorem, the joint distribution of outputs for any collection of inputs is multivariate Gaussian, as in a GP, and completely characterized by some kernel function $\kappa(\cdot, \cdot)$. Let $\sigma(\cdot)$ be some non-linear (activation) function, such as the ReLU or sigmoid, and $\mathbf{w}$ and $b$ be the network weights and biases. Then, the associated kernel for the infinite-width network can be formulated in terms of [38, 12]:

$$\kappa(\mathbf{x}, \mathbf{x}') = \int p(\mathbf{w}) \, p(b) \, \sigma(\mathbf{w}^\mathsf{T} \mathbf{x} + b) \, \sigma(\mathbf{w}^\mathsf{T} \mathbf{x}' + b) \, \mathrm{d}\mathbf{w} \, \mathrm{d}b, \tag{1}$$

where priors $p(\mathbf{w})$ and $p(b)$ are chosen suitably. From the formulation in Eq. (1) one can read that the covariance function corresponding to an untrained (random) single-layer neural network is *fully characterized* by the choice of activation function $\sigma(\cdot)$ and the priors on the network weights and biases.

Many of the covariance functions corresponding to commonly used activation functions under Gaussian priors on the weights have closed-form representations. The ArcCos kernel [3] covers the ReLU (ArcCos-1) and the step (ArcCos-0) activations, the so-called 'neural network kernel' [59, 44] links to sigmoidal activations. Gram matrices $\mathbf{K}_{ij} = \kappa(x_i, x'_j)$, evaluated for input pairs $x$ and $x'$ over $x, x' \in [-3, 3]$, are visualized in Fig. 2a to illustrate these covariance structures. The models induced by the ReLU, step, and sigmoid activations are by nature local, with the concentration of the non-linearity around the origin. As can be seen in the 1D regression results in Fig. 2b, the strong inductive bias in the prior (model) is reflected in the posterior. The ReLU extrapolates by matching the local bias of the data, while the step and sigmoidal saturate outside the data, and priors are clearly sensitive to translations and *highly non-stationary*.

**Stationarity** For a *stationary* (homogeneous) covariance function the covariance structure of the model functions $f(\mathbf{x})$ is the same regardless of the absolute position in the input space, and thus the parametrization can be relaxed to $\kappa(\mathbf{x}, \mathbf{x}') \triangleq \kappa(\mathbf{x} - \mathbf{x}') = \kappa(\mathbf{r})$. Stationarity implies translation-invariance, *i.e.* the notion of similarity between inputs $\mathbf{x}$, and $\mathbf{x}'$ is only a function of their distance $\mathbf{x} - \mathbf{x}'$. This implies *weak stationarity* under stochastic process theory [40], and results in *reversion to the prior* outside observed data (see Fig. 2b). For stationary covariance functions, the best-known family is the Matérn [32, 49] family of kernels, which features models with sample functions of varying degrees of differentiability (smoothness):

$$\kappa_{\mathrm{Mat.}}(\mathbf{x}, \mathbf{x}') = \frac{2^{1-\nu}}{\Gamma(\nu)} \left( \sqrt{2\nu} \, \frac{\|\mathbf{x} - \mathbf{x}'\|}{\ell} \right)^\nu \mathrm{K}_\nu \left( \sqrt{2\nu} \, \frac{\|\mathbf{x} - \mathbf{x}'\|}{\ell} \right), \tag{2}$$

where $\nu$ is a smoothness parameter, $\ell$ a characteristic length-scale parameter, $\mathrm{K}_\nu(\cdot)$ the modified Bessel function, and $\Gamma(\cdot)$ the gamma function. For the Matérn class, the processes are $\lceil \nu \rceil - 1$ times mean-square differentiable, and the family includes the RBF (squared exponential) and the exponential (Ornstein–Uhlenbeck) kernels as limiting cases as $\nu \to \infty$ and $\nu = 1/2$.

**Local stationarity** Neural networks do not naturally exhibit stationarity. Formally, this stems from the problem of representing non-linearities over an infinite input domain with a finite set of local non-linear mappings. However, to bridge the gap between neural networks and widely-used stationary kernels, the typical approach is to restrict the invariance to be *local* (see Fig. 2a). This is the approach in the RBF-NN [60, 44] and general Matérn-NN [34] activation functions, where the prior is a composite covariance function with a Gaussian decay envelope (see discussion in [15]), *i.e.*,

$$\kappa_{\mathrm{Mat\text{-}NN}}(\mathbf{x}, \mathbf{x}') \propto \exp(-\mathbf{x}^\mathsf{T}\mathbf{x}/2\sigma_{\mathrm{m}}^2) \, \kappa_{\mathrm{Mat.}}(\mathbf{x}, \mathbf{x}') \exp(-\mathbf{x}'^\mathsf{T}\mathbf{x}'/2\sigma_{\mathrm{m}}^2), \tag{3}$$

where $\sigma_{\mathrm{m}}^2 = 2\sigma_{\mathrm{b}}^2 + \ell^2$ (see Fig. 2a for the decay effect of the envelope). One motivation for this approach is to assume a Gaussian input density on the training/testing inputs, thus restricting our

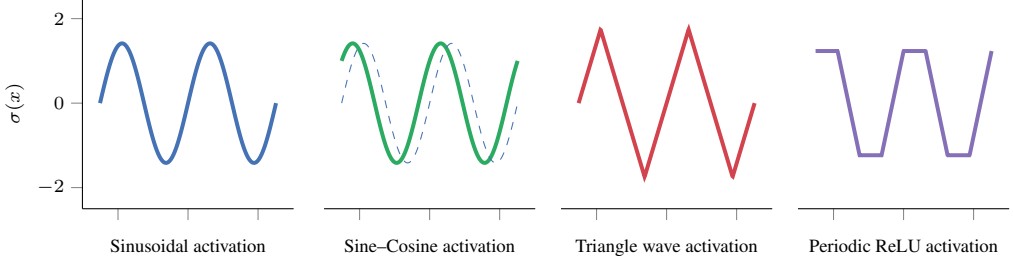

Figure 3: Sketches of types of periodic activations functions $\sigma(\cdot)$ visualized over two repeating periods each. The two leftmost are infinitely differentiable and the two rightmost are piecewise linear. The dashed line overlays the Sinusoidal activation for comparison.

interest only to inputs in the vicinity of the training data. The local behaviour is highlighted in Fig. 2b, where the locally stationary models revert to the mean (as expected), but the marginal uncertainty of the model functions drops to zero when far enough from training samples. This motivates us to seek activation functions that induce (globally) stationary behaviour (bottom-rows in Fig. 2).

## 3 Global Stationarity-inducing Activation Functions

We derive a direct connection between (fully) stationary covariance functions and periodic activation functions. Our main result leverages the spectral duality of the spectral density of stationary covariance functions to establish a direct connection between the weight prior and the induced prior on the function space. Furthermore, we show that this connection is not restricted to the (sinusoidal) Fourier basis and establishes an equivalence relationship between a Student-$t$ distribution on the weights and a prior on the function space with Matérn covariance structure.

### 3.1 Spectral Duality Under Harmonic Functions

For stationarity-inducing priors, the covariance function can be written equivalently in terms of its spectral density function. This stems from *Bochner's theorem* (see [6] and 36A in [28] for a more general formulation) which states a bounded continuous positive definite function $\kappa(\mathbf{r})$ can be represented through the following norm-preserving isomorphism

$$\kappa(\mathbf{r}) = \int \omega(\mathbf{r})\,\mu(\mathrm{d}\boldsymbol{\omega}), \tag{4}$$

where $\mu$ is a positive measure and $\mathbf{r} \in \mathbb{R}^d$. If the measure $\mu(\boldsymbol{\omega})$ has a density, it is called the *spectral density* $S(\boldsymbol{\omega})$ corresponding to the covariance function $\kappa(\mathbf{r})$. With $\omega(\mathbf{r}) = (2\pi)^{-d}\exp(\mathrm{i}\,\boldsymbol{\omega}^{\mathsf{T}}\mathbf{r})$, this gives rise to the Fourier duality of covariance and spectral density, which is known as the *Wiener–Khinchin theorem* (*e.g.*, [44]):

$$\kappa(\mathbf{r}) = \frac{1}{(2\pi)^d}\int S(\boldsymbol{\omega})\,e^{\mathrm{i}\,\boldsymbol{\omega}^{\mathsf{T}}\mathbf{r}}\,\mathrm{d}\boldsymbol{\omega} \quad\text{and}\quad S(\boldsymbol{\omega}) = \int \kappa(\mathbf{r})\,e^{-\mathrm{i}\,\boldsymbol{\omega}^{\mathsf{T}}\mathbf{r}}\,\mathrm{d}\mathbf{r}. \tag{5}$$

For $d > 1$, if the covariance function is *isotropic* (it only depends on the Euclidean norm $\|\mathbf{r}\|$ such that $\kappa(r) \triangleq \kappa(\|\mathbf{r}\|)$), then it is invariant to all rigid motions of the input space. Moreover, the spectral density will also only depend on the norm of the dual input variable $\boldsymbol{\omega}$. In the neural network case, we can assume the previous layer will take care of scaling the inputs, and thus we effectively are interested in the isotropic case, which brings us to analysing 1D projections.

An important question is whether we are restricted to the (sinusoidal) Fourier basis. Let $\psi_j(x)$ be any sequence of bounded, square-integrable (over its period) periodic functions, such that these functions define an inner product space (generalising a basis of a vector space). In signal processing terms, they define a *frame* that provides a redundant yet stable way of representing a signal. From a generalised harmonic perspective [58], we could resort to any convenient basis, such as Gabor wavelets [11]. However, in this work we seek to choose $\psi(x)$ suitably, such that we have uniform convergence to $\omega(r)$ (*cf.*, Eq. (4)), ensuring that we retain the spectral density $S(\omega)$.

## 3.2 Types of Periodic Activation Functions

We show how various periodic activation functions (see Fig. 4) under the view of Eq. (1) directly link back to the spectral duality. These periodic activation functions are continuous, bounded, and centered at zero, while not necessarily differentiable and never monotonic over $\mathbb{R}$ (but monotonic within half-period).

**Sinusoidal activation**  By assuming the activation function to be a sinusoid, $\sigma_{\sin}(x) = \sqrt{2}\sin(x)$ (see Fig. 4), placing a uniform prior on the bias term, $p(b) = \mathrm{Uniform}(-\pi, \pi)$, and substituting this into Eq. (1), we can obtain an expression for the covariance function $\kappa(x, x')$:

$$\kappa(x, x') = \int p(w) \int_{-\pi}^{\pi} \frac{1}{\pi} \sin(wx + b) \sin(wx' + b) \, \mathrm{d}b \, \mathrm{d}w. \tag{6}$$

By solving the inner integral and applying Euler's formula ($\cos(z) = \frac{1}{2}(e^{\mathrm{i}z} + e^{-\mathrm{i}z})$), we obtain:

$$\kappa(x, x') = \frac{1}{2} \int p(w) e^{\mathrm{i}w(x-x')} \, \mathrm{d}w + \frac{1}{2} \int p(w) e^{-\mathrm{i}w(x-x')} \, \mathrm{d}w, \tag{7}$$

which simplifies under the assumption that the prior on the weight is symmetric and has support on the real line, *i.e.*,

$$\kappa(x, x') = \int p(w) e^{\mathrm{i}w(x-x')} \, \mathrm{d}w. \tag{8}$$

By letting $r = x - x'$, we find that we recover the spectral density decomposition of a stationary process given by the Wiener–Khinchin theorem in Eq. (5), where $p(w) = \frac{1}{2\pi}S(w)$. Therefore, we obtain a direct connection between the prior on the weights and the spectral density. A detailed derivation can be found in App. A.1.

**Sine–Cosine activation**  An alternative way of using sinusoidal activation functions is the use of a composite sine–cosine activation, *i.e.*, $\sigma_{\sin\cos}(x) = \sin(x) + \cos(x)$ (Fig. 4 compares the sin–cos to the sinusoidal activation). Using such construction has the benefit of removing the need to integrate over the bias, which can result in reduced variance [51] of the estimates, giving us a covariance function in the form of:

$$\kappa(x, x') = \int p(w) \left[\sin(wx) + \cos(wx)\right] \left[\sin(wx') + \cos(wx')\right] \, \mathrm{d}w$$

$$= \int p(w) \sin(w(x + x')) \, \mathrm{d}w + \int p(w) \cos(w(x - x')) \, \mathrm{d}w. \tag{9}$$

By application of Euler's formula and under the usual assumption that $p(w)$ is symmetric and has support on the entire real line, the above reduces to:

$$\kappa(x, x') = \frac{1}{2} \left[\int p(w) e^{\mathrm{i}w(x-x')} \, \mathrm{d}w + \int p(w) e^{-\mathrm{i}w(x-x')} \, \mathrm{d}w\right] = \int p(w) e^{\mathrm{i}w(x-x')} \, \mathrm{d}w. \tag{10}$$

Again, by letting $r = x - x'$, we find that we recover the spectral density decomposition of a stationary process given by the Wiener–Khinchin theorem. A detailed derivation can be found in App. A.2.

**Triangle wave activation**  Inspired by the success of piecewise linear activation functions (*e.g.*, ReLU [36], leaky ReLU [54], PReLU [17]), we show that the triangle wave, a periodic piecewise linear function, can be used instead of sinusoidal activations. The triangle wave is given by:

$$\psi(x) = \frac{4}{p} \left(x - \frac{p}{2} \left\lfloor \frac{2x}{p} + \frac{1}{2} \right\rfloor\right) (-1)^{\lfloor \frac{2x}{p} + \frac{1}{2} \rfloor}, \tag{11}$$

where $p$ is the period and $\lfloor \bullet \rfloor$ denotes the floor function. By considering a period of $p = 2\pi$, assuming a uniform prior on the biases, $b \sim \mathrm{Uniform}(-\pi, \pi)$, and appropriate rescaling, we get the activation

$$\sigma_{\mathrm{triangle}}(x) = \frac{\pi}{2\sqrt{2}} \left(x - \pi \left\lfloor \frac{x}{\pi} + \frac{1}{2} \right\rfloor\right) (-1)^{\lfloor \frac{x}{\pi} + \frac{1}{2} \rfloor}, \tag{12}$$

for which we obtain a direct correspondence of the network weight priors to the spectral density. As shown in the derivations in App. A.3, we find that we again recover the spectral density decomposition of a stationary process given by the Wiener–Khinchin theorem.

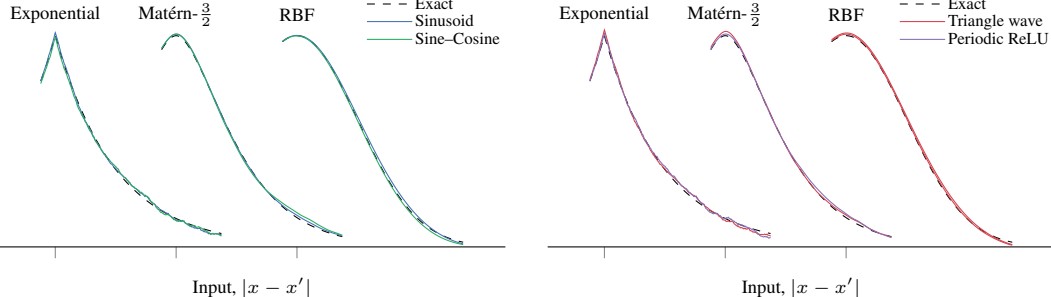

Figure 4: Covariance functions calculated by MC integration with 5000 samples and compared to their exact dashed counterparts (peaks shifted for clarity). Regardless of the type of periodic activation function used, we can recover the behaviour of the stationary kernels (see also Fig. 8a).

**Periodic ReLU activation**   To go even further, we define a piecewise linear periodic activation function with a repeating rectified linear pattern, which we call the periodic ReLU function. It can be defined in terms of the sum of two triangle waves, with the second one being shifted by ¼ of a period. The resulting periodic function is piecewise linear and defined as:

$$\psi(x) = \frac{8}{\pi^2}\left(\left((x+\frac{\pi}{2})-\pi\left\lfloor\frac{(x+\frac{\pi}{2})}{\pi}+\frac{1}{2}\right\rfloor\right)(-1)^{\lfloor\frac{(x+\frac{\pi}{2})}{\pi}+\frac{1}{2}\rfloor} + \left(x-\pi\left\lfloor\frac{x}{\pi}+\frac{1}{2}\right\rfloor\right)(-1)^{\lfloor\frac{x}{\pi}+\frac{1}{2}\rfloor}\right),$$
(13)

assuming a period of $p = 2\pi$. It is again possible to obtain a correspondence between the network weight prior and the spectral density by considering a weighted version of the equation above. In particular, we show in App. A.4 that from the following weighted periodic ReLU activation function,

$$\sigma_{\text{pReLU}}(x) = \frac{\pi}{4}\left(\left((x+\frac{\pi}{2})-\pi\left\lfloor\frac{(x+\frac{\pi}{2})}{\pi}+\frac{1}{2}\right\rfloor\right)(-1)^{\lfloor\frac{(x+\frac{\pi}{2})}{\pi}+\frac{1}{2}\rfloor} + \left(x-\pi\left\lfloor\frac{x}{\pi}+\frac{1}{2}\right\rfloor\right)(-1)^{\lfloor\frac{x}{\pi}+\frac{1}{2}\rfloor}\right),$$
(14)

we again recover the spectral density decomposition of a stationary process given by the Wiener–Khinchin theorem, providing a direct connection between the prior on the weights and the spectral density. Note that, choosing a piecewise linear periodic activation function has potential computational benefits compared to sinusoidal activation functions and can help prevent vanishing gradients.

## 3.3   Kernel Functions

We have established that is it possible to obtain a direct correspondence of the prior on the weights and the spectral density of a stationary covariance function by using periodic activation functions in random neural networks. In App. B we show that by placing a Student-$t$ distribution on the weights with degrees of freedom of $u = 2\nu$ we recover the spectral density of the Matérn family, *i.e.*,

$$p(w) = \frac{\Gamma(\frac{u+1}{2})}{\sqrt{u\pi}\Gamma(\frac{u}{2})}\left(1+\frac{w^2}{u}\right)^{-\frac{u+1}{2}} = \frac{1}{2\pi}2\sqrt{\pi}\frac{\Gamma(\nu+\frac{1}{2})}{\Gamma(\nu)}(2\nu)^\nu\left(2\nu+w^2\right)^{-(\nu+\frac{1}{2})} = \frac{1}{2\pi}S_{\text{Mat.}}(w),$$
(15)

where $p(w)$ denotes the probability density function of a Student-$t$ distribution and $S_{\text{Mat.}}(w)$ denotes the spectral density of the Matérn family. This means that a Student-$t$ prior on the weights in combination with an appropriately scaled periodic activation function corresponds directly to a prior in the function space with Matérn covariance structure. Fig. 4 verifies this result and shows that we recover the exact covariance function (dashed) for various examples from the Matérn family by Monte Carlo (MC) integration (5000 samples) with all of the discussed periodic activation functions.

Recall that the Matérn family contains the exponential kernel ($\nu = 1/2$) and the RBF Kernel ($\nu = \infty$) as special cases. Similarly, the Student-$t$ distribution has the Cauchy ($u = 1$) and the Normal distribution ($u = \infty$) as limiting cases, resulting in correspondence to the respective special cases in the Matérn family. Table 2 in App. B summarises the set of priors on the weights that correspond to the spectral density of kernel functions in the Matérn family. This correspondence is exact for the sinusoidal and sine–cosine activations. For the triangle wave and periodic ReLU activations, the obtained correspondence is approximate, and the introduced approximation error is analyzed in detail in App. A.

Table 1: Examples of UCI regression tasks, showing the globally stationary NN model directly gives competitive RMSE and for most data sets also improved mean negative log predictive density (NLPD) compared to non-stationary and locally stationary NN models. KFAC Laplace was used as the inference method. High std is due to variability in 10-fold-cv splits of the small data sets, not method performance.

| $(n, d)$
$(c, n_{\text{batch}})$ | BOSTON
(506, 12)
(1, 50) | | CONCRETE
(1030, 5)
(1, 50) | | AIRFOIL
(1503, 5)
(1, 50) | | ELEVATORS
(16599, 18)
(1, 500) | |
|---|---|---|---|---|---|---|---|---|
| | NLPD | RMSE | NLPD | RMSE | NLPD | RMSE | NLPD | RMSE |
| ReLU | 0.51±0.32 | 0.37±0.07 | 0.78±0.16 | 0.48±0.04 | 0.51±0.53 | 0.41±0.21 | 0.38±0.03 | 0.35±0.01 |
| loc RBF | 0.52±0.30 | 0.37±0.08 | 0.78±0.22 | 0.44±0.05 | 0.10±0.15 | 0.26±0.03 | 0.41±0.04 | **0.35±0.01** |
| glob RBF (sin) | 0.42±0.34 | 0.36±0.07 | 0.74±0.15 | 0.49±0.05 | 0.14±0.17 | 0.29±0.05 | 0.38±0.03 | 0.35±0.01 |
| glob RBF (prelu) | 0.39±0.30 | **0.36±0.07** | 0.74±0.14 | 0.49±0.04 | **0.05±0.12** | **0.26±0.03** | 0.74±0.73 | 0.46±0.21 |
| loc Mat-3/2 | 0.71±0.38 | 0.40±0.08 | 0.84±0.28 | **0.42±0.04** | 0.11±0.18 | 0.26±0.03 | 0.43±0.04 | 0.35±0.01 |
| glob Mat-3/2 (sin) | 0.43±0.27 | 0.39±0.08 | 0.73±0.16 | 0.49±0.05 | 0.07±0.15 | 0.27±0.03 | **0.37±0.03** | 0.35±0.01 |
| glob Mat-3/2 (prelu) | **0.38±0.22** | 0.38±0.08 | **0.72±0.18** | 0.48±0.05 | 0.08±0.12 | 0.27±0.03 | 0.39±0.03 | 0.36±0.01 |

## 4 Experiments

To assess the performance of stationarity-inducing activation functions, we compared the in-domain and out-of-domain predictive uncertainty to non-stationary and locally stationary models on various benchmark data sets. In all experiments, the compared models differ only by the activation function and the respective weight priors of the last hidden layer of the network. Replacing the prior effects the initialization of the NN weights and the training objective, as we maximize the log joint density. A detailed description of the experimental setup is available in App. D. The illustrative toy BNN examples are implemented using Turing.jl [14], GP regression results use GPflow [33], and all other experiments are implemented using PyTorch [41].

**Illustrative toy examples**  Fig. 1 shows predictive densities for non-stationary, locally stationary, and globally stationary activation functions on the banana classification task. The top row illustrates the predictive densities of infinite-width BNNs (GP), the bottom row shows corresponding results for a single hidden layer BNN using 30 hidden units. We observe that global stationarity-inducing activation functions revert to the prior outside the data, leading to conservative behaviour (high uncertainty) for out-of-domain samples. Moreover, we see that the finite-width BNNs result in similar behaviour to their infinite-width counterpart, while locally stationary activation functions in finite-width BNNs exhibit a slower reversion to the mean than their infinite-width corresponding GPs. Additionally, we include a 1D toy regression study that highlights the differences between different prior assumptions encoded by choice of the activation function. Fig. 2 shows the corresponding prior covariance as well as posterior predictions for the infinite-width (GP) model. We replicated the experiment with a finite-width network and observed that the behaviour translates to finite width (see Fig. 8 in App. D.1).

**UCI benchmarks**  Table 1 shows results on UCI [8] regression data sets comparing deep neural networks with ReLU, locally stationary RBF [60], and locally stationary Matérn-³/₂ [34] against global stationary models. An extended results table can be found in App. D.2 and additional results

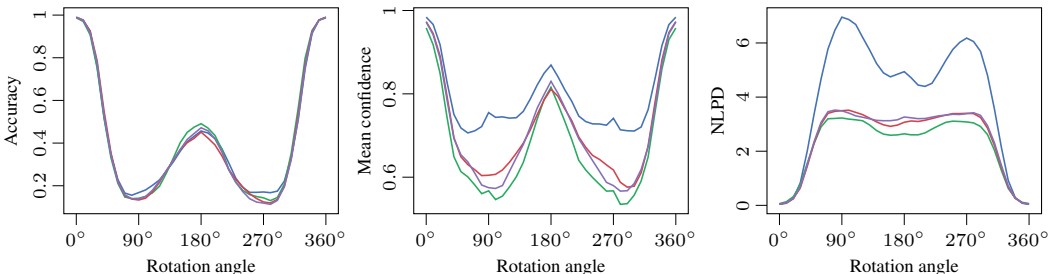

Figure 5: Rotated MNIST: The models have been trained on unrotated digits. The test-set digits are rotated at test time to show the sensitivity of the trained model to perturbations. All models perform equally in terms of accuracy, while ReLU (——) shows overconfidence in terms of mean confidence and NLPD. The stationary RBF models (—— local, —— sin, —— sin–cos) capture uncertainty.

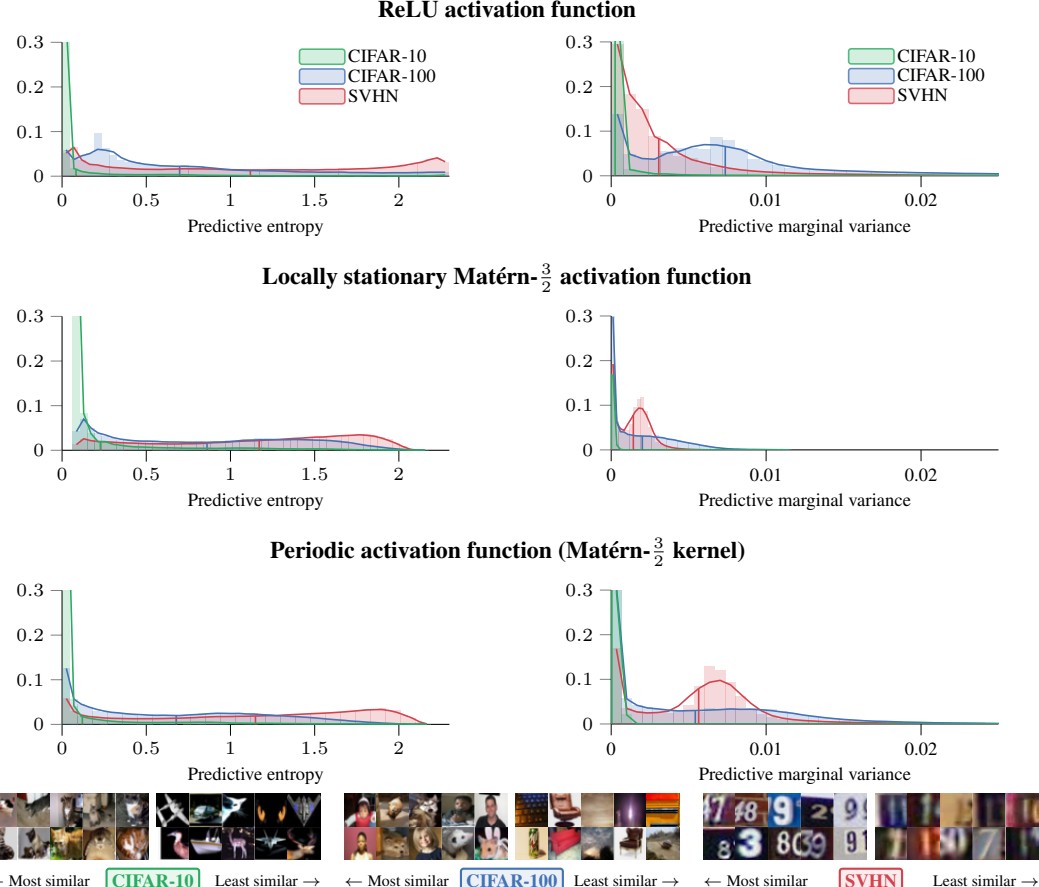

Figure 6: OOD detection experiment results for models trained on CIFAR-10 and tested on CIFAR-10, CIFAR-100, and SVHN. Top rows show predictive entropy and predictive marginal variance histograms of test images for in-domain and OOD data sets. Top to bottom: ReLU activation function, local stationary Matérn-$3/2$ activation function [34], and sinusoidal activation function that induces a globally stationary Matérn-$3/2$ model. The last row shows sample images from each test set: left-side images with the lowest entropy/highest confidence and right-side images with the highest entropy/lowest confidence. See App. D.5 for histograms for more models.

on UCI classification tasks in App. D.3. Table 1 lists root mean square error (RMSE) and negative log predictive density (NLPD), which captures the predictive uncertainty, while the RMSE only accounts for the mean. Global stationary models provide better estimates of the target distribution in all cases while obtaining comparable RMSEs. It is important to note that the large standard deviations in Table 1 are due to the small number of data points and the fact that some splits in the 10-fold CV end up being harder than others. The posterior inference is performed using KFAC Laplace [45].

**Detection of distribution shift with rotated MNIST** Fig. 5 demonstrates the predictive behaviour of different activation functions under covariate shift. We evaluate the predictive accuracy, mean confidence of the predicted class, and NLPD on MNIST [25] for different rotations of the input. Models are trained on unrotated digits using KFAC Laplace [45]. The results indicate that all models obtain similar accuracy results, while only local and global stationary models do not result in over-confident uncertainty estimates. For an ideally calibrated model, the mean confidence would decrease as low as the accuracy curve when the digits are rotated, keeping the NLPD values as low as possible.

**Out-of-distribution detection using CIFAR-10, CIFAR-100, and SVHN** Fig. 6 shows model performance on OOD detection in image classification for ReLU, locally stationary Matérn-$3/2$, and globally stationary Matérn-$3/2$ models. The models have been trained using SWAG [29] on CIFAR-10 [23], and tested on CIFAR-10, CIFAR-100 [23], and SVHN [39] test set images. Both CIFAR-100 (more similar) and SVHN (more dissimilar to CIFAR-10) images are OOD data, and the

models should show high uncertainties for the respective test images. We use two metrics to measure model uncertainty: predictive entropy and predictive marginal variance. Predictive entropy captures uncertainty present in the predicted mean class weights, while the predictive marginal variance captures uncertainty related to the variability in the predicted class weights. From these two metrics, the predictive marginal variance could be more suitable as an OOD detection metric, as predictive entropy can justifiably give high values for some in-distribution samples, for example, those that the model confidently predicts being in between two known classes.

The histograms of predictive entropies for different test sets show that all models can separate between in-distribution and OOD data to some extent, and all models consider SVHN more OOD than CIFAR-100, which is intuitive as CIFAR-100 resembles CIFAR-10 more. Also, the predictive marginal variance histograms show that the models have higher variance on OOD data compared to in-distribution data. However, the ReLU model considers CIFAR-100 more OOD than SVHN and the locally stationary Matérn-$3/2$ model has overall relatively small predictive marginal variance values for all data sets. We consider predictive marginal variance a better metric for OOD detection than predictive entropy, as in-distribution samples that are hard to classify are expected to have high predictive entropy, not necessarily allowing the detection of OOD samples on this metric. See Fig. 14 in App. D.5 for results on different models, and Table 8 in App. D.5 for numerical results showing the area under the receiver operating characteristic curve and the area under the precision-recall curve for the OOD detection experiment.

## 5 Discussion and Conclusions

We have shown that *periodic* activation functions in Bayesian neural networks (BNNs) establish a direct connection between the prior on the network weights and the spectral density of the limiting stationary process. This link goes beyond the Fourier (sinusoidal) basis, which we have illustrated by deriving the correspondence also for the triangle wave and a new periodic variant of the ReLU activation function. Moreover, we have shown that for BNNs with a periodic activation function, placing a Student-$t$ distribution on the network weights corresponds to a prior in the function space with Matérn covariance structure. This correspondence is exact for the sinusoidal and sin–cos activations and approximate for the triangle and periodic ReLU activations.

Our work can help build neural network models that are more robust and less vulnerable to OOD data—occurring either naturally or maliciously. As demonstrated in the experiments, the modelling assumptions have practical importance in reducing sensitivity to input shift. Perturbation of inputs can be captured in the last layer of a deep model, and the principled choice of *a priori* stationarity has shown to be effective based on the experiments. Using periodic activation functions in BNNs induces *global* stationarity, *i.e.*, the model is not only locally translation-invariant, which induces conservative behaviour across the input domain. The resulting model reverts to the prior for out-of-distribution (OOD) samples, resulting in well-specified uncertainties.

In contrast to models with non-stationary or locally stationary activations, optimising and performing approximate inference in models with periodic activation functions has proven to be more challenging. In particular, we found that the uncertainty over the bias terms is often not accurately estimated when using KFAC Laplace or SWAG, which required us to use a relatively large number of hidden units. However, the experiments using dynamic HMC indicate that the number of hidden units can be drastically reduced using a more accurate approximate inference scheme, as seen in Fig. 1 and Fig. 11 in App. D.1.

The codes and data to replicate the results are available under MIT license at `https://github.com/AaltoML/PeriodicBNN`.

## Acknowledgments and Disclosure of Funding

We acknowledge the computational resources provided by the Aalto Science-IT project. This research was supported by the Academy of Finland grants 324345 and 339730, and Saab Finland Oy. We thank William J. Wilkinson and the anonymous reviewers for feedback on the manuscript.

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
