# Supplementary Material:
# Periodic Activation Functions Induce Stationarity

This supplementary document is organized as follows. App. A includes further details and derivations for the periodic activation functions in the Methods section of the main paper. App. B provides a full derivation of the correspondence between the Student-$t$ prior and Matérn family under periodic activation functions. App. C presents and discusses further links to existing work. App. D includes details on the experiments, baseline methods, data sets, and additional tables and result plots.

## A  Derivations for Activation Functions

In this section we will introduce the detailed derivations for the activation functions presented in the main text. In each derivation, we start from the definition of the covariance function of a random single-layer neural network given as:

$$\kappa(x, x') = \int \int p(w) \, p(b) \, \sigma(wx + b)\sigma(wx' + b) \, \mathrm{d}b \, \mathrm{d}w, \qquad (16)$$

where $\sigma(\cdot)$ denotes the activation function and $p(w)$ and $p(b)$ the priors over weights and biases respectively.

### A.1  Sinusoidal Activation Function

Let the activation function $\sigma(\cdot)$ and prior $p(b)$ take the following form:

$$\sigma(x) = \sqrt{2} \sin(x), \qquad (17)$$
$$p(b) = \mathrm{Uniform}(-\pi, \pi). \qquad (18)$$

Then the covariance function $\kappa(x, x')$ from Eq. (16) can now be written as:

$$\kappa(x, x') = \int p(w) \int_{-\pi}^{\pi} \frac{1}{\pi} \sin(wx + b) \sin(wx' + b) \, \mathrm{d}b \, \mathrm{d}w. \qquad (19)$$

The inner integral (over $b$) can now be solved by considering that $\sin(x)\sin(y) = \frac{1}{2}[\cos(x - y) - \cos(x + y)]$, i.e.,

$$\int_{-\pi}^{\pi} \sin(wx + b) \sin(wx' + b) \, \mathrm{d}b$$
$$= \int_{-\pi}^{\pi} \frac{\cos(wx + b - wx' - b) - \cos(wx + b + wx' + b)}{2} \, \mathrm{d}b$$
$$= \frac{1}{2} \int_{-\pi}^{\pi} \cos(w(x - x')) - \cos(w(x + x') + 2b) \, \mathrm{d}b$$
$$= \pi \cos(w(x - x')). \qquad (20)$$

Plugging the above back into Eq. (19) gives us:

$$\kappa(x, x') = \int p(w) \cos(w(x - x')) \, \mathrm{d}w, \qquad (21)$$

which, by application of Euler's formula $\cos(z) = \frac{1}{2}(e^{\mathrm{i}z} + e^{-\mathrm{i}z})$, can be written as:

$$\kappa(x, x') = \frac{1}{2} \int p(w)e^{\mathrm{i}w(x-x')} \, \mathrm{d}w + \frac{1}{2} \int p(w)e^{-\mathrm{i}w(x-x')} \, \mathrm{d}w. \qquad (22)$$

Since the integration is over the entire real line we can do a change of variables $w = -w$ in the second integral and by assuming a symmetric prior on $w$, we obtain

$$\kappa(x, x') = \frac{1}{2} \int p(w) e^{iw(x-x')} \, dw + \frac{1}{2} \int p(-w) e^{iw(x-x')} \, dw$$

$$= \int p(w) e^{iw(x-x')} \, dw. \tag{23}$$

By letting $r = x - x'$, we find that we recover the spectral density decomposition of a stationary process given by the Wiener–Khinchin theorem, *i.e.*,

$$\kappa(r) = \int p(w) \, e^{iwr} \, dw, \tag{24}$$

where $p(w) = \frac{1}{2\pi} S(w)$, which was to be shown.

## A.2 Sine–Cosine Activation Function

Let us assume a bias-free random single-layer neural network. Following Eq. (16), in the infinite limit, the corresponding covariance function would take the form:

$$\kappa(x, x') = \int p(w) \, \sigma(wx) \sigma(wx') \, dw. \tag{25}$$

Let the activation function be given as:

$$\sigma(x) = \sin(x) + \cos(x). \tag{26}$$

This gives the covariance function in the form

$$\kappa(x, x') = \int p(w) \, [\sin(wx) + \cos(wx)] \, [\sin(wx') + \cos(wx')] \, dw$$

$$= \int p(w) \, \sin(w(x + x')) \, dw + \int p(w) \, \cos(w(x - x')) \, dw. \tag{27}$$

By application of Euler's formula we get

$$\kappa(x, x') = \frac{1}{2} \Big[ \int p(w) \, e^{iw(x-x')} \, dw + \int p(w) \, e^{-iw(x-x')} \, dw$$

$$+ \int p(w) \, i e^{iw(x+x')} \, dw - \int p(w) \, i e^{-iw(x+x')} \, dw \Big], \tag{28}$$

where under assumption that $p(w)$ is symmetric and has support on the entire real line the above reduces to:

$$\kappa(x, x') = \frac{1}{2} \left[ \int p(w) \, e^{iw(x-x')} \, dw + \int p(w) \, e^{-iw(x-x')} \, dw \right]$$

$$= \int p(w) \, e^{iw(x-x')} \, dw, \tag{29}$$

where the last step follows from application of the change of variables and the symmetricity of $p(w)$. Again, by letting $r = x - x'$, we find that we recover the spectral density decomposition of a stationary process given by the Wiener–Khinchin theorem, *i.e.*,

$$\kappa(r) = \int p(w) \, e^{iwr} \, dw \tag{30}$$

confirming the statement in the main paper.

## A.3 Triangle Wave Activation

The triangle wave is a periodic, piecewise linear function and it can be written in a parametric form as

$$\psi(x) = \frac{4}{p} \left( x - \frac{p}{2} \left\lfloor \frac{2x}{p} + \frac{1}{2} \right\rfloor \right) (-1)^{\lfloor \frac{2x}{p} + \frac{1}{2} \rfloor}, \tag{31}$$

where $p$ is the period of the triangle wave. In the following derivation we will assume that $p = 2\pi$. For analysis, we note that the Fourier series approximation to $\psi(x)$ can be given in the following form (in the limit of $n$):

$$\psi(x) = \lim_{n \to \infty} \frac{8}{\pi^2} \sum_{k=0}^{n-1} (-1)^k (2k+1)^{-2} \sin((2k+1)x). \tag{32}$$

Let $\lambda_k := 2k + 1$, and let us assume the triangle wave activation function and a uniform prior on $b$ (as in the preceding derivation):

$$\sigma(z) = \sqrt{2} \sum_{k=0}^{n-1} (-1)^k \lambda_k^{-2} \sin(\lambda_k z), \tag{33}$$

$$p(b) = \text{Uniform}(-\pi, \pi), \tag{34}$$

where $n$ is the number of harmonics to include in the approximation and $k$ is the harmonic label.

The covariance function $\kappa(x, x')$ is, therefore, given as:

$$\kappa(x, x') = \int p(w) \int_{-\pi}^{\pi} 2p(b) \left[ \sum_{k=0}^{n-1} (-1)^k \lambda_k^{-2} \sin(\lambda_k(wx+b)) \right]$$
$$\left[ \sum_{j=0}^{n-1} (-1)^j \lambda_j^{-2} \sin(\lambda_j(wx'+b)) \right] \mathrm{d}w \, \mathrm{d}b. \tag{35}$$

Now, let us solve the inner integrals by ignoring the constant terms and assuming that $k \neq j$:

$$\int_{-\pi}^{\pi} \frac{1}{\pi} \sin(\lambda_k(wx+b)) \sin(\lambda_j(wx'+b)) \, \mathrm{d}b$$
$$= \int_{-\pi}^{\pi} \frac{1}{2\pi} \cos(\lambda_k(wx+b) - \lambda_j(wx'+b)) - \cos(\lambda_k(wx+b) + \lambda_j(wx'+b)) \, \mathrm{d}b. \tag{36}$$

By solving the definite integral above, we obtain:

$$= \frac{\sin(w(\lambda_k x - \lambda_j x')) - \sin(w(\lambda_k x - \lambda_j x')) + 2\pi(\lambda_k - \lambda_j))}{\lambda_j - \lambda_k}$$
$$+ \frac{\sin(w(\lambda_k x + \lambda_j x')) - \sin(w(\lambda_k x + \lambda_j x') + 2\pi(\lambda_k + \lambda_j))}{\lambda_j + \lambda_k}, \tag{37}$$

where we recognise that $2\pi(\lambda_k - \lambda_j)$ will always be an even multiple of $2\pi$, as both $\lambda_k$ and $\lambda_j$ are odd. Thus, the above cancels out and Eq. (35) reduces to containing only indexes $k = j$:

$$\kappa(x, x') = \int p(w) \int_{-\pi}^{\pi} 2p(b) \sum_{k=0}^{n-1} \frac{(-1)^{2k}}{\lambda_k^4} \sin(\lambda_k(wx+b)) \sin(\lambda_k(wx'+b)) \, \mathrm{d}w \, \mathrm{d}b. \tag{38}$$

The solution to the first integral for every summand is given as:

$$\int_{-\pi}^{\pi} 2p(b) \sin(\lambda_k(wx+b)) \sin(\lambda_k(wx'+b)) \, \mathrm{d}b$$
$$= \int_{-\pi}^{\pi} \frac{1}{2\pi} \cos(w\lambda_k(x - x')) - \cos(w\lambda_k(x + y) + 2b\lambda_k) \, \mathrm{d}b$$
$$= \cos(w\lambda_k(x - x')) - \underbrace{\frac{\cos(w\lambda_k(x + x') + 2\lambda_k\pi) \sin(2\lambda_k\pi))}{\lambda_k}}_{=0}. \tag{39}$$

Inserting this into the equation of the covariance function results in:

$$\kappa(x, x') = \int p(w) \sum_{k=0}^{n-1} \frac{(-1)^{2k}}{\lambda_k^4} \cos(w\lambda_k(x - x')) \, dw. \tag{40}$$

We now take into account that the exact solution is given when $n \to \infty$ and that $(-1)^{2k} = 1$:

$$\kappa(x, x') = \int \lim_{n\to\infty} p(w) \sum_{k=0}^{n-1} \frac{1}{\lambda_k^4} \cos(w\lambda_k(x - x')) \, dw. \tag{41}$$

Here we can use the dominated convergence theorem to take the limit outside of the integral. For this we have $f_n(w) = p(w) \sum_{k=0}^{n-1} \frac{1}{\lambda_k^4} \cos(w\lambda_k(x - x'))$, and we need a function $g(w)$ for which $|f_n(w)| \le g(w)$ for all $n$ and $\int g(w) \, dw < \infty$:

$$|f_n(w)| = p(w) \sum_{k=0}^{n-1} \frac{1}{(2k+1)^4} |\cos(w\lambda_k(x - x'))| \tag{42}$$

$$\le p(w) \sum_{k=0}^{n-1} \frac{1}{(2k+1)^4} \tag{43}$$

$$\le p(w) \sum_{k=1}^{\infty} \frac{1}{k^4} = \frac{\pi^4}{90} p(w) \tag{44}$$

Hence, by choosing $g(w) = \frac{\pi^4}{90} p(w)$ the requirements to use the dominated convergence theorem are satisfied since $\int \frac{\pi^4}{90} p(w) \, dw = \frac{\pi^4}{90} < \infty$.

We obtain:

$$\kappa(x, x') = \lim_{n\to\infty} \int p(w) \sum_{k=0}^{n-1} \frac{1}{\lambda_k^4} \cos(w\lambda_k(x - x')) \, dw. \tag{45}$$

It is also possible to write the density on the weights in Eq. (45) in the form of a mixture density (assuming the density $p$ is in the location-scale family, which is the case for the prior distributions discussed in this paper):

$$\kappa(x, x') = \lim_{n\to\infty} \int \sum_{k=0}^{n-1} \frac{1}{\lambda_k^4} p(w) e^{i\lambda_k w(x-x')} \, dw,$$

$$\kappa(x, x') = \lim_{n\to\infty} \int \sum_{k=0}^{n-1} \pi_k p(w) e^{i\lambda_k w(x-x')} \, dw, \tag{46}$$

$$\kappa(x, x') = \lim_{n\to\infty} \int \sum_{k=0}^{n-1} \pi_k p(w \,|\, \lambda_k) e^{iw(x-x')} \, dw,$$

where $p(w \,|\, \lambda_k)$ denotes the density function of $p(w)$ with scale parameter $\lambda_k$. Let us now denote $\lim_{n\to\infty} \sum_{k=0}^{n-1} \pi_k p(w \,|\, \lambda_k)$ as $\hat{p}(w)$, *i.e.*,

$$\kappa(x, x') = \int \hat{p}(w) e^{iw(x-x')} \, dw. \tag{47}$$

By letting $r = x - x'$, we find that we again recover the spectral density decomposition of a stationary process given by the Wiener–Khinchin theorem, *i.e.*,

$$\kappa(r) = \int \hat{p}(w) \, e^{iwr} \, dw, \tag{48}$$

which recovers the statement in the main paper.

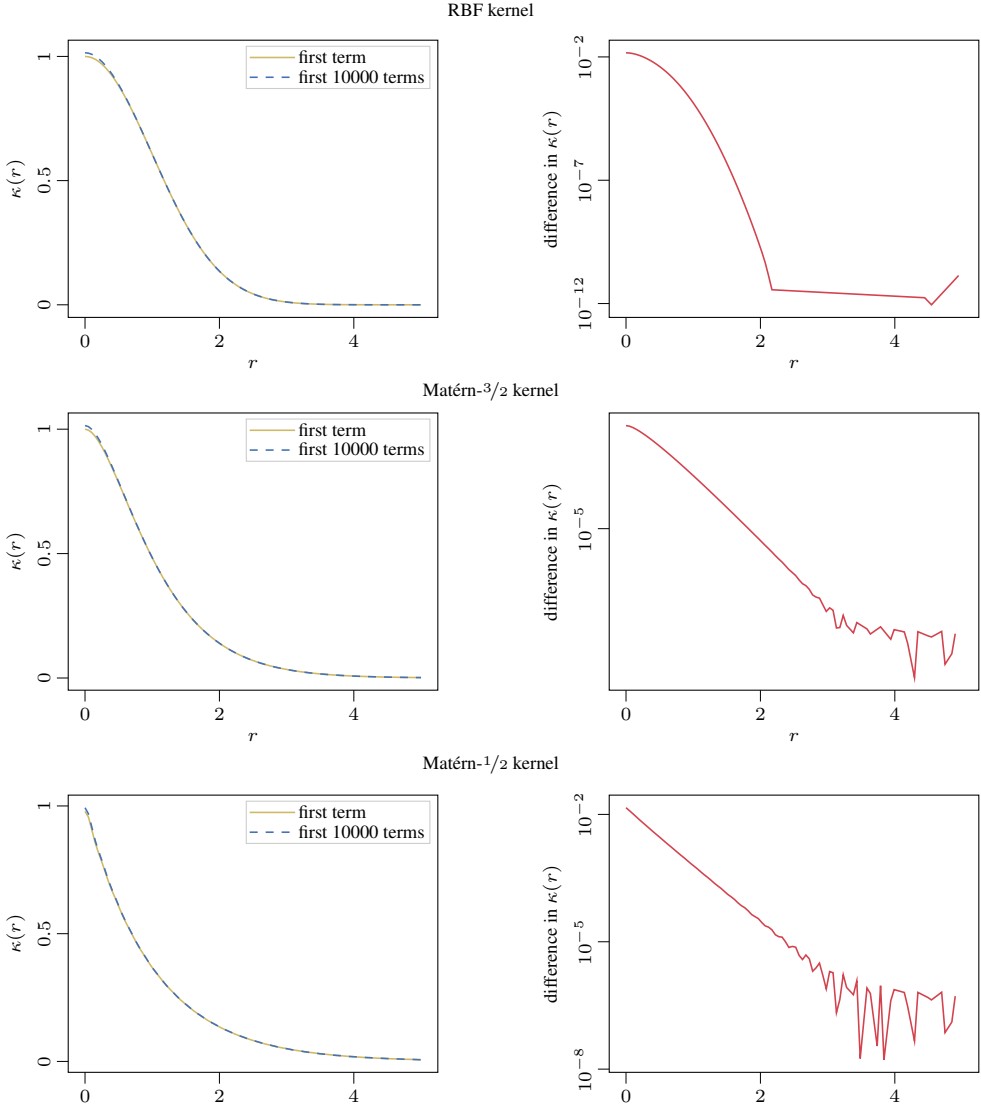

Figure 7: Numerical integration results on the error introduced by approximating the mixture in Eq. (48) using only its first component. The top row figures correspond to the RBF kernel, with $p(w)$ being standard normal distribution. The middle and bottom row figures correspond to the Matérn-$3/2$ and Matérn-$1/2$ kernels, with $p(w)$ being Student-t distribution with degrees of freedom 3 and 1 respectively. The figures on the left compare the kernel values $\kappa(r)$ from equation Eq. (45) with $r = x - x'$, while the sum has either only the first term or the 100 first terms included. The figures on the right show the difference between the curves on the left side figures on a logarithmic scale. The figures show that the error introduced by approximating the mixture in Eq. (48) using only its first component is very small when $p(w)$ is one of the distributions used in the experiments.

We again obtain a connection between the prior (in this case, a mixture) and the spectral density through the Wiener–Khinchin theorem. In this case, the spectral density also has to admit a mixture density. However, working with a mixture density as prior could be potentially challenging for inference. For this reason, in practise the mixture is approximated only using its first component, since this is a series of rapidly decreasing terms. Fig. 7 shows simulation results of the error introduced by approximating the mixture using only its first component. Based on these results the error from this approximation appears very small, when $p(w)$ is in the Matérn-class, which is the case in all experiments in this paper.

## A.4 Periodic ReLU Activation

Finally, we consider a periodic function that captures the spirit of the ReLU activation function in the form of a repeating rectified linear pattern. This can be modelled as the sum of two triangle waves, the second one shifted by half a period. The resulting periodic function is piece-wise linear and defined as:

$$\psi(x) = \frac{2}{\pi}\left(\left((x+\frac{\pi}{2})-\pi\left\lfloor\frac{(x+\frac{\pi}{2})}{\pi}+\frac{1}{2}\right\rfloor\right)(-1)^{\lfloor\frac{(x+\frac{\pi}{2})}{\pi}+\frac{1}{2}\rfloor} + \left(x-\pi\left\lfloor\frac{x}{\pi}+\frac{1}{2}\right\rfloor\right)(-1)^{\lfloor\frac{x}{\pi}+\frac{1}{2}\rfloor}\right), \quad (49)$$

assuming a period of $p = 2\pi$. In the following derivation we will again use a Fourier series approximation of a slightly differently weighted form of the periodic ReLU function as the activation function, *i.e.*,

$$\sigma(x) = \lim_{n\to\infty}\sum_{k=0}^{n-1}(-1)^k\lambda_k^{-2}\left(\sin(\lambda_k(x+\frac{\pi}{2})) + \sin(\lambda_k x)\right), \quad (50)$$

and $p(b) = \mathrm{Uniform}(-\pi, \pi)$. Then, the covariance function is given as:

$$\kappa(x, x') = \int p(w)\int p(b)\left[\lim_{n\to\infty}\sum_{k=0}^{n-1}(-1)^k\lambda_k^{-2}\left(\sin(\lambda_k(wx+\frac{\pi}{2}+b)) + \sin(\lambda_k(wx+b))\right)\right]$$
$$\left[\lim_{n\to\infty}\sum_{j=0}^{n-1}(-1)^j\lambda_j^{-2}\left(\sin(\lambda_j(wx'+\frac{\pi}{2}+b)) + \sin(\lambda_j(wx'+b))\right)\right]\,\mathrm{d}w\,\mathrm{d}b. \quad (51)$$

As done previously, we will first solve the inner integral by assuming that $k \neq j$, *i.e.*,

$$\int p(b)\frac{(-1)^{k+j}}{\lambda_k^2\lambda_j^2}\left(\sin(\lambda_k(wx+\frac{\pi}{2}+b)) + \sin(\lambda_k(wx+b))\right)$$
$$\left(\sin(\lambda_j(wx'+\frac{\pi}{2}+b)) + \sin(\lambda_j(wx'+b))\right)\,\mathrm{d}b, \quad (52)$$

where we recognise that,

$$\sin(\lambda_k(wx+\frac{\pi}{2}+b)) = (-1)^k\cos(\lambda_k(wx+b)), \quad (53)$$

due to the definition of $\lambda_k$, giving us a series of definite integrals:

$$\int_{-\pi}^{\pi}(-1)^{k+j}\cos(\lambda_k(x+b))\cos(\lambda_j(x'+b))\,\mathrm{d}b$$
$$= \frac{(-1)^{k+j}}{(\lambda_j-\lambda_k)(\lambda_j+\lambda_k)}\Big(2\sin(\pi\lambda_j)\cos(\pi\lambda_k)(\lambda_k\sin(\lambda_j x')\sin(\lambda_k x) + \lambda_j\cos(\lambda_j x')\cos(\lambda_k x))$$
$$- 2\cos(\pi\lambda_j)\sin(\pi\lambda_k)(\lambda_j\sin(\lambda_j x')\sin(\lambda_k x) + \lambda_k\cos(\lambda_j x')\cos(\lambda_k x))\Big)$$
$$= 0, \quad (54)$$

which cancels out as $\sin(\pi\lambda_k)$ and $\sin(\pi\lambda_j)$ will be zero for every $j$ and every $k$. Moreover, we have:

$$\int_{-\pi}^{\pi}\sin(\lambda_k(x+b))\sin(\lambda_j(x'+b))\,\mathrm{d}b$$
$$= \frac{1}{(\lambda_j-\lambda_k)(\lambda_j+\lambda_k)}\Big(2\sin(\pi\lambda_j)\cos(\pi\lambda_k)(\lambda_j\sin(\lambda_j x')\sin(\lambda_k x) + \lambda_k\cos(\lambda_j x')\cos(\lambda_k x))$$
$$- 2\cos(\pi\lambda_j)\sin(\pi\lambda_k)(\lambda_k\sin(\lambda_j x')\sin(\lambda_k x) + \lambda_j\cos(\lambda_j x')\cos(\lambda_k x))\Big)$$
$$= 0, \quad (55)$$

which again cancels out for the same reason. Finally, we have:

$$\int_{-\pi}^{\pi} (-1)^j \sin(\lambda_k(x+b)) \cos(\lambda_j(x'+b)) \, \mathrm{d}b$$

$$= \frac{(-1)^j}{(\lambda_j - \lambda_k)(\lambda_j + \lambda_k)} \Big( 2(\cos(\pi\lambda_j)\sin(\pi\lambda_k)(\lambda_j \sin(\lambda_j x')\cos(\lambda_k x) - \lambda_k \cos(\lambda_j x')\sin(\lambda_k x))$$

$$+ \sin(\pi\lambda_j)\cos(\pi\lambda_k)(\lambda_j \cos(\lambda_j x')\sin(\lambda_k x) - \lambda_k \sin(\lambda_j x')\cos(\lambda_k x))) \Big)$$

$$= 0, \tag{56}$$

and the same can be shown for $\int_{-\pi}^{\pi} (-1)^k \sin(\lambda_j(x'+b)) \cos(k(x+b)) \, \mathrm{d}b$.

Therefore, our derivations simplify to containing only terms where $k = j$:

$$\kappa(x, x') = \int p(w) \int p(b) \left[ \lim_{n \to \infty} \sum_{k=0}^{n-1} (-1)^{2k} \lambda_k^{-4} \left( (-1)^k \cos(\lambda_k(wx+b)) + \sin(\lambda_k(wx+b)) \right) \right.$$

$$\left. \left( (-1)^k \cos(\lambda_k(wx'+b)) + \sin(\lambda_k(wx'+b)) \right) \right] \mathrm{d}w \, \mathrm{d}b. \tag{57}$$

Here we can again use the dominated convergence theorem to take the limit outside of the integrals:

$$\kappa(x, x') = \lim_{n \to \infty} \int p(w) \sum_{k=0}^{n-1} (-1)^{2k} \lambda_k^{-4} \int p(b) \left[ \left( (-1)^k \cos(\lambda_k(wx+b)) + \sin(\lambda_k(wx+b)) \right) \right.$$

$$\left. \left( (-1)^k \cos(\lambda_k(wx'+b)) + \sin(\lambda_k(wx'+b)) \right) \right] \mathrm{d}w \, \mathrm{d}b. \tag{58}$$

We can now start by calculating the inner integral:

$$\int p(b) \left[ \left( (-1)^k \cos(\lambda_k(wx+b)) + \sin(\lambda_k(wx+b)) \right) \right.$$

$$\left. \left( (-1)^k \cos(\lambda_k(wx'+b)) + \sin(\lambda_k(wx'+b)) \right) \right] \mathrm{d}b$$

$$= \int p(b) \left[ \cos(\lambda_k(wx+b)) \cos(\lambda_k(wx'+b)) + \sin(\lambda_k(wx+b)) \sin(\lambda_k(wx'+b)) \right.$$

$$+ (-1)^k \cos(\lambda_k(wx+b)) \sin(\lambda_k(wx'+b))$$

$$\left. + (-1)^k \cos(\lambda_k(wx'+b)) \sin(\lambda_k(wx+b)) \right] \mathrm{d}b$$

$$= \int p(b) \left[ \cos(\lambda_k(wx+b) - \lambda_k(wx'+b)) + (-1)^k \sin(\lambda_k(wx+b) + \lambda_k(wx'+b)) \right] \mathrm{d}b$$

$$= \int p(b) \cos(\lambda_k w(x-x') \, \mathrm{d}b + (-1)^k \underbrace{\int p(b) \sin(\lambda_k w(x+x') + 2\lambda_k b)) \, \mathrm{d}b}_{=0}$$

$$= \cos(\lambda_k w(x - x')) \tag{59}$$

Inserting this result into the original equation results in:

$$\kappa(x, x') = \lim_{n \to \infty} \int p(w) \sum_{k=0}^{n-1} \lambda_k^{-4} \cos(w\lambda_k(x - x')) \, \mathrm{d}w, \tag{60}$$

where we recognise that this the exact same equation encountered in App. A.3 and the terms in the series decrease rapidly towards zero, thus, allowing us to approximate the covariance using only the

Table 2: Priors on the weights corresponding to the spectral density of kernels in the Matérn family.

| KERNEL FUNCTION | PRIOR DISTRIBUTION |
|---|---|
| Exponential ($\nu = 1/2$) | $\mathrm{Cauchy}(0, 1)$ |
| Matérn-$\nu$ | t-dist$(2\nu)$ |
| RBF ($\nu \to \infty$) | $\mathrm{N}(0, 1)$ |

first term of the sum which gives us (see App. A.3 for analysis on the approximation error):

$$\kappa(x, x') = \int p(w) \cos(w(x - x')) \, \mathrm{d}w = \int p(w) \, e^{iw(x - x')} \, \mathrm{d}w. \tag{61}$$

Finally, by letting $r = x - x'$, we find that we again recover the spectral density decomposition of a stationary process given by the Wiener–Khinchin theorem, *i.e.*,

$$\kappa(r) = \int p(w) \, e^{iwr} \, \mathrm{d}w, \tag{62}$$

which concludes the derivation.

## B  Derivations of Correspondence to the Matérn family

Relating to Sec. 3.3 in the main paper, we provide the following derivation. The spectral density of the Matérn family (*cf.*, Eq. (2)) for the 1D case is given as:

$$S_{\mathrm{Mat.}}(w) = 2\sqrt{\pi} \frac{\Gamma(\nu + \frac{1}{2})}{\Gamma(\nu)} (2\nu)^{\nu} \left(2\nu + w^2\right)^{-(\nu + \frac{1}{2})}, \tag{63}$$

where $\nu$ is a smoothness parameter, and we assume unit magnitude and $\ell = 1$ for simplicity. Note that we intentionally used $w$ instead of $\omega$ to highlight the correspondence to the prior on the weights. By assuming the prior $p(w)$ to follow a Student-$t$ distribution *i.e.*,

$$p(w) = \frac{\Gamma(\frac{u+1}{2})}{\sqrt{u\pi}\Gamma(\frac{u}{2})} \left(1 + \frac{w^2}{u}\right)^{-\frac{u+1}{2}}, \tag{64}$$

and setting the degree of freedom $u = 2\nu$, we can recover the spectral density of the Matérn family, *i.e.*,

$$
\begin{aligned}
p(w) &= \frac{\Gamma(\frac{2\nu+1}{2})}{\sqrt{2\nu\pi}\Gamma(\frac{2\nu}{2})} \left(1 + \frac{w^2}{2\nu}\right)^{-\frac{2\nu+1}{2}} &&= \frac{\Gamma(\nu + \frac{1}{2})}{\sqrt{2\nu\pi}\Gamma(\nu)} \left(\frac{1}{2\nu}(2\nu + w^2)\right)^{-(\nu + \frac{1}{2})} \\
&= \frac{\Gamma(\nu + \frac{1}{2})}{\sqrt{2\nu\pi}\Gamma(\nu)} (2\nu)^{\nu + \frac{1}{2}} \left(2\nu + w^2\right)^{-(\nu + \frac{1}{2})} &&= \frac{\Gamma(\nu + \frac{1}{2})}{\sqrt{\pi}\Gamma(\nu)} (2\nu)^{\nu} \left(2\nu + w^2\right)^{-(\nu + \frac{1}{2})} \\
&= \frac{1}{2\pi} 2\sqrt{\pi} \frac{\Gamma(\nu + \frac{1}{2})}{\Gamma(\nu)} (2\nu)^{\nu} \left(2\nu + w^2\right)^{-(\nu + \frac{1}{2})} &&= \frac{1}{2\pi} S_{\mathrm{Mat.}}(w). \tag{65}
\end{aligned}
$$

A summary of the priors on the network weights corresponding to the spectral density of kernels in the Matérn family is given in Table 2.

## C  Additional Insights

Fourier features, the sinusoidal (or other periodic) basis, and the special dual relationship between stationary covariance functions and the associated spectral density, have been common building blocks in machine learning and signal processing methods for decades. We review connections to Random Fourier features and Fourier methods in GP models.

## C.1 Connection to Random Fourier Features

Random Fourier features [43] are a popular technique for randomized, low-dimensional approximations of kernel functions. They are motivated by the observation that the spectral density of a RBF covariance function of a Gaussian process prior can be estimated using Monte Carlo integration. Let $\boldsymbol{\omega} \sim \mathrm{N}(0,1)$ be Gaussian distributed and $\zeta_{\boldsymbol{\omega}}(\mathbf{x}) = \exp\left(\mathrm{i}\,\boldsymbol{\omega}^\mathsf{T}\mathbf{x}\right)$, then

$$\mathbb{E}_{\boldsymbol{\omega}}[\zeta_{\boldsymbol{\omega}}(\mathbf{x}), \zeta_{\boldsymbol{\omega}}(\mathbf{x}')^*] = \int p(\boldsymbol{\omega})\,\exp\left(\mathrm{i}\,\boldsymbol{\omega}^\mathsf{T}\mathbf{r}\right)\,\mathrm{d}\boldsymbol{\omega}, \tag{66}$$

where $^*$ denotes the complex conjugate, is an estimator of the covariance function.

Assuming that $\boldsymbol{\omega}$ and $\mathbf{x}$ are real-valued, let $b$ be a value drawn uniformly from $[-\pi, \pi]$, and by replacing the complex exponentials with cosines $z_{\boldsymbol{\omega}}(\mathbf{x}) = \sqrt{2}\cos(\boldsymbol{\omega}^\mathsf{T}\mathbf{x} + b)$ we obtain:

$$
\begin{aligned}
\mathbb{E}_{\boldsymbol{\omega}}[z_{\boldsymbol{\omega}}(\mathbf{x})z_{\boldsymbol{\omega}}(\mathbf{x}')] &= \mathbb{E}_{\boldsymbol{\omega}}[\cos(\boldsymbol{\omega}^\mathsf{T}(\mathbf{x} - \mathbf{x}'))] \\
&= \mathbb{E}_{\boldsymbol{\omega}}[\mathbb{E}_b[\sqrt{2}\cos(\boldsymbol{\omega}^\mathsf{T}\mathbf{x} + b)\,\sqrt{2}\cos(\boldsymbol{\omega}^\mathsf{T}\mathbf{x}' + b) \mid \boldsymbol{\omega}]] \quad \text{(Euler's formula)} \\
&\approx \frac{1}{K}\sum_{j=1}^{K} \sqrt{2}\cos(\boldsymbol{\omega}_k^\mathsf{T}\mathbf{x} + b_k)\,\sqrt{2}\cos(\boldsymbol{\omega}_k^\mathsf{T}\mathbf{x}' + b_k).
\end{aligned}
\tag{67}
$$

We recognise that the estimate used in random Fourier features is similar to the covariance function of a finite-width single hidden layer network with sinusoidal activation function.

Rahimi and Recht [43] additionally proposed a representation using a composition of a sin and a cosine function, which motivated the use of the sine–cosine activation function in this paper. Sutherland and Schneider [51] later showed that this representation reduces the variance of the estimates when used to approximate the RBF kernel.

## C.2 Fourier Methods in Gaussian Process Models

The Fourier duality for stationary covariance functions has been extensively leveraged in Gaussian process models. For gridded inputs, this duality directly allows for leveraging FFT methods to speed up inference and learning. In particular, the sparse spectrum GP (SSGP) method [24] uses the spectral representation of the covariance function to draw random samples from the spectrum. These samples are used to represent the GP on a trigonometric basis, *i.e.*,

$$\phi(\mathbf{x}) = \left(\cos(2\pi\,\mathbf{s}_1^\top\mathbf{x})\,\sin(2\pi\,\mathbf{s}_1^\top\mathbf{x})\,\ldots\cos(2\pi\,\mathbf{s}_h^\top\mathbf{x})\,\sin(2\pi\,\mathbf{s}_h^\top\mathbf{x})\right), \tag{68}$$

where the spectral points $\mathbf{s}_r, r = 1, 2, \ldots, h\ (2h = m)$ are sampled from the spectral density of the stationary covariance function (following the normalization convention used in the original paper). The covariance function corresponding to the SSGP can be given in the form (*cf.*, Mercer's theorem):

$$\kappa(\mathbf{x}, \mathbf{x}') \approx \frac{2}{m}\,\phi(\mathbf{x})\,\phi^\top(\mathbf{x}') = \frac{1}{h}\sum_{r=1}^{h} \cos\left(2\pi\,\mathbf{s}_r^\top(\mathbf{x} - \mathbf{x}')\right). \tag{69}$$

This representation of the sparse spectrum method converges to the full GP in the limit of the number of spectral points going to infinity, and is the preferred formulation of the method in one or two dimensions (discussed in [24]). We can interpret the SSGP method in Eq. (69) as a Monte Carlo approximation of the Wiener–Khinchin integral. This interpretation also gives rise to alternative methods for GPs: the methods by Hensman et al. [19] and Solin and Särkkä [48] can be interpreted as a dense/structural (quadrature) approximation to the same integral.

However, for high-dimensional inputs, the SSGP method requires optimization of the frequencies rather than relying on sampling, which is problematic (as discussed in [24]), resulting in a tendency to overfit, and loses the interpretation of the original GP prior in the model. Note that these issues have been addressed in subsequent work. As in SSGPs, our method can be seen as a sampling/optimization approach to a rank-reduced approximation of the induced prior covariance structure. However, the connection we derived retains the role of the prior throughout and generalizes the interpretation of the role of the periodic basis.

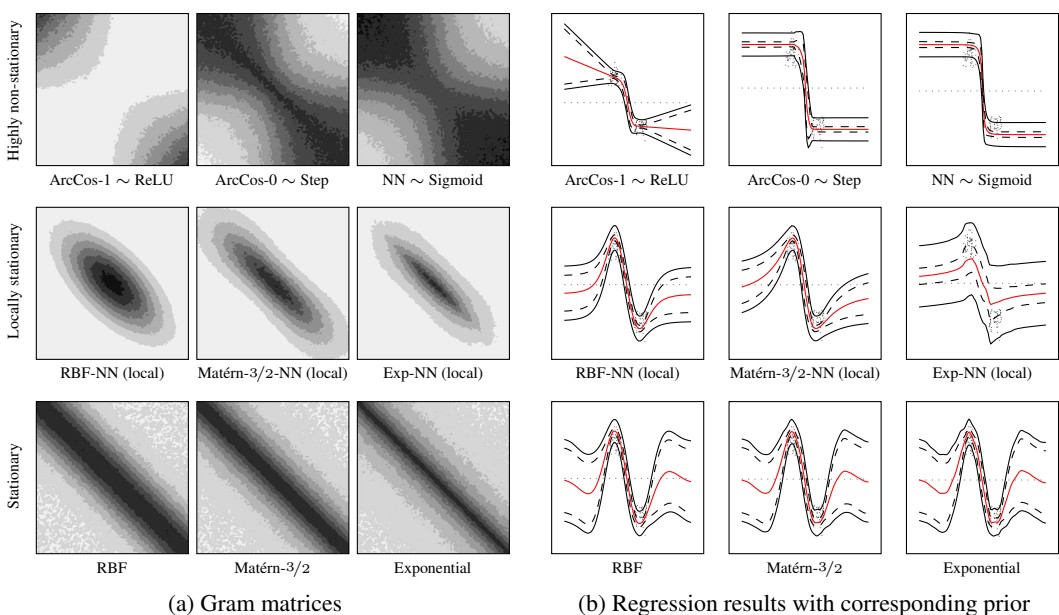

(a) Gram matrices       (b) Regression results with corresponding prior

Figure 8: **Left:** Gram matrices (evaluated for $\hat{\kappa}(x, x')$ with 1000 Monte Carlo samples) corresponding to the prior covariance induced by different finite-width NNs (10 hidden units). **Right:** 1D regression results corresponding to the model induced by the prior in the left-hand panels, showing the posterior and predictive 95% intervals of a BNN with ten hidden units obtained through sampling with dynamic HMC for 5000 iterations.

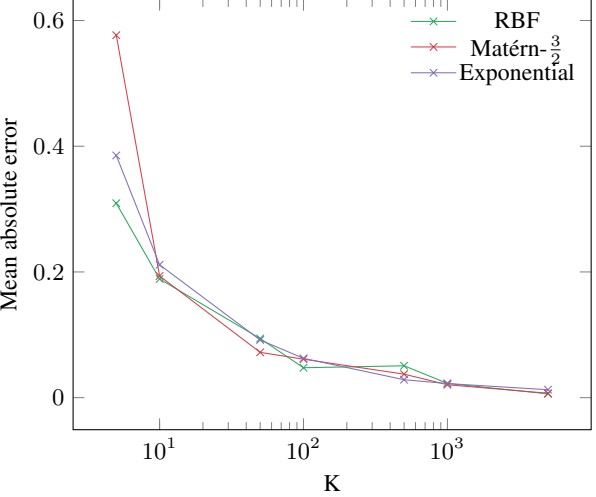

Figure 9: Simulation estimates of the error between the Gram matrix of the limiting process and the Gram matrix of a finite width model with sinusoidal activation functions under increasing number of hidden units ($K$).

# D    Experiment Details and Additional Results

Our main contribution is linking globally stationary GP priors to a corresponding infinite-width NN with one hidden layer by specifying activation functions and priors on weights and biases. Most applications, however, require NNs with a deeper structure. In the experiments, this is achieved by considering the preceding NN structure as a feature extractor. The layer containing the model specification is added as the last hidden fully connected layer after the feature extractor part. This means that the activation function and priors for weights and biases only apply to the last hidden layer of the full NN architecture. It is important to note here that we do not mean the final output

layer that produces the class weights or regression output when we refer to the last hidden layer, but the linear layer preceding the output layer. For NN architectures that would not otherwise have a fully connected hidden layer before the output layer, one such layer is added. We refer to this last hidden layer that contains the model implementation as the 'model layer'. The weights and biases in the model layer are initialized from the prior distributions defined by the model. Table 2 lists prior distributions corresponding to different globally stationary priors. Prior distributions for locally stationary and non-stationary ReLU models are $N(0, 1)$ following the results from [34]. Many of these priors require using periodic activation functions instead of the ReLU, which could be expected to affect trainability. In our experiments, we observed that sometimes training neural networks with periodic activation functions can be a bit harder compared to ReLU networks, but usually, slightly adjusting the learning rate is enough to solve these issues. We expect that the issues with trainability were minor as the periodic activation functions are only used in a single layer of the NN, while other parts of the NN are still using ReLU.

Although our theoretical result considers an infinitely wide NN, we observed in practice that a good result can be obtained if the number of hidden units is sufficiently larger than the dimensionality of the preceding feature space. To avoid using an excessively large number of hidden units, we have added a bottleneck layer for NN architectures for which the feature space after the feature-extraction part would have high dimensionality. This way, we could use a model layer with roughly 100 times more hidden units than the number of dimensions in the preceding feature space.

In the derivations for the prior on the weights, we expected $\ell = 1$ for simplicity. It may be necessary for the model to have a different lengthscale depending on the data in practice. We implemented this by multiplying the weights by a lengthscale parameter before multiplying the input by these weights in the forward pass. This way, we do not need to adjust the prior distribution for the weights as the weights themselves will still stay in the space corresponding to $\ell = 1$. The lengthscale parameter is added as a trainable parameter with a prior of $\ell \sim \Gamma(\alpha = 2, \beta = 0.5)$, which is weakly informative.

The model layer produces hidden features which are mapped to outputs using the linear output layer. The weights for the output layer are initialized from $N(0, 1/\kappa)$. The output $f(x)$ dimensionality equals the number of classes in classification tasks and one in regression tasks. Based on this output, the data likelihood needs to be calculated for the loss function. For regression tasks the data likelihood is $N(y - f(x) \,|\, 0, s^2)$, where $s$ is the standard deviation of the measurement noise which we include into the model as a trainable parameter with prior $\Gamma(\alpha = 0.5, \beta = 1)$. For classification, the data likelihood is calculated by applying the softmax function on $f(x)$ to map the outputs to probabilities and then choosing the class probability corresponding to the class $y$.

To train the model, we construct a loss function that considers both the data likelihood and the prior distributions. Since our goal is to fit an approximate posterior distribution on the model parameters, we use the Bayes formula to obtain a loss function that is directly proportional to the posterior distribution:

$$p(w, b, \ell, s \,|\, y, x) \propto p(y \,|\, x)\, p(w)\, p(b)\, p(\ell)\, p(s). \tag{70}$$

For optimization purposes we take a negative logarithm of the product of priors and data likelihood, which also changes this to a minimization problem, resulting in the following loss function:

$$\mathcal{L} = -\log p(y \,|\, x) - \log p(w) - \log p(b) - \log p(\ell) - \log p(s). \tag{71}$$

Here $p(y \,|\, x)$ is the data likelihood described above (for classification the cross entropy loss directly gives $-\log p(y|x)$ from the outputs $f(x)$), $p(w)$ is the prior on weights, $p(b)$ is the prior on biases, $p(\ell)$ is the prior on lengthscale, and $p(s)$ is the prior on measurement noise variance (for classification $-\log p(s) = 0$). For NN architectures that have a feature-extractor part preceding the model layer, we also include standard L2-regularization on the parameters of the feature-extractor network to prevent the model from completely bypassing the defined priors by learning extreme values for feature-extractor network parameters. For globally stationary models $p(b) = \text{Uniform}(-\pi, \pi)$, meaning that the bias term is defined on a constrained space. We therefore optimise $\hat{b} \in \mathbb{R}$, which is defined on an unconstrained space, and apply the map/link function $b = 2\pi\, \text{sigmoid}(\hat{b}) - \pi$.

The calculations for obtaining the experiment results were mostly performed using computer resources within the Aalto Science-IT project. These resources included both CPU nodes and GPU nodes (NVIDIA V100 and Tesla P100). Some results were also calculated on local GPU resources (NVIDIA RTX 2080). All of the utilized data sets are publicly available and widely used, and none of them contain any personally identifiable information or offensive content. The illustrative toy BNN

examples are implemented using HMC in Turing.jl [14], GP regression results use GPflow [33], and all other experiments are implemented using PyTorch [41].

## D.1 Illustrative Toy Examples

The NN model architecture for the illustrative toy examples contains only the model layer with 30 hidden units and the output layer. The posterior estimates are obtained through dynamic HMC sampling [14] run for 10k iterations and four independent chains. Fig. 1 shows predictive densities for non-stationary, locally stationary, and globally stationary activation functions on the banana classification task. The top row illustrates the predictive densities of infinite-width BNNs (GP), and the bottom row shows corresponding results for a finite-width BNN. We observe that models with global stationarity-inducing activation functions revert to the prior outside the data, leading to conservative behaviour (high uncertainty) for out-of-domain samples. Moreover, we see that the finite-width BNNs result in similar behaviour to their infinite-width counterpart, while the locally stationary activation functions in finite-width BNNs exhibit a slower reversion to the mean than their infinite-width corresponding GPs. Fig. 10 shows additional BNN results for the same experiment for globally stationary models using different periodic activation functions. We see that we obtain similar results regardless of the choice of the periodic activation function. As expected, all periodic activation functions result in low variance only for the training data clusters and revert to the prior outside the data. Fig. 11 shows the effect of varying the number of hidden units in the same experiment.

Additionally, we include a 1D toy regression study highlighting the differences between different prior assumptions encoded by choice of the activation function. Fig. 2 shows the corresponding prior covariance as well as posterior predictions for the infinite-width (GP) model. In Fig. 8, we replicate the same study with a finite-width network and recover the same behaviour. For the finite-width results, posterior estimates are obtained through dynamic HMC sampling for 5000 iterations. Fig. 9 illustrates the error between the Gram matrices of the infinite width and finite width models.

## D.2 Benchmark Regression Tasks

For the UCI [8] regression tasks, the NN architecture is a fully connected network with layers $d$-1000-1000-500-25-2000-1. A dropout layer with $p = 0.1$ is applied at the 500 nodes wide layer to prevent overfitting. For these experiments, we used a 10-fold cross-validation setup performed for a single repetition per experiment. Each model is trained for 100 epochs using SGD (momentum 0.9). The batch sizes used for each data set are listed in Table 3. The learning rates for the lengthscale parameter $\ell$ and measurement noise standard deviation $s$ were set to 0.01. All learning rates were decreased to 0.72 of the original value during training, using a schedule having square root dependence on the progression through epochs (slower than linear learning rate decay). The lengthscale parameter $\ell$ was initialized with a value of five, and $s$ was initialized with one. For the non-stationary ReLU model, the lengthscale parameter does not have a similar significance as in the global and local stationary models and therefore was initialized with a value of one to prevent vanishing or exploding gradients. Posterior inference was performed using KFAC Laplace [45].

Since we observed that the results were sensitive to the SGD learning rate and the variance scale parameter of KFAC Laplace, we performed a grid search over both of these hyperparameters. Therefore, we split the training set such that 80% is used for training and the remaining 20% is used as a validation set for the grid search. First, a grid search for the SGD learning rate was performed over the values $[5 \times 10^{-5}, 1 \times 10^{-4}, 5 \times 10^{-4}, 1 \times 10^{-3}]$, choosing the learning rate that achieved the smallest RMSE on the validation set. A single common learning rate was chosen for all ten folds. Subsequently, KFAC Laplace was applied on models trained using the best learning rate, using variance scales in the grid $[0.01, 0.05, 0.1, 0.15, 0.2, 0.25]$. 30 samples were used for model averaging in the grid search. As the objective in the KFAC Laplace variance scale grid search, we used a weighted sum of validation set data negative log-likelihood and an OOD noise validation set negative log-likelihood [22]. We used $\lambda = 0.2$ as the parameter controlling the weighting between the regular validation set score and the OOD noise set score (one-fifth of the weight on the OOD set). A single common KFAC Laplace variance scale was chosen for all ten folds. After the best learning rate and KFAC Laplace variance scale has been selected, the model is retrained from the start on the full training set of each fold using the best learning rate. KFAC Laplace model with the best variance scale is then fitted on the trained model, and 50 samples from the approximate posterior are used for model averaging to obtain the final results.

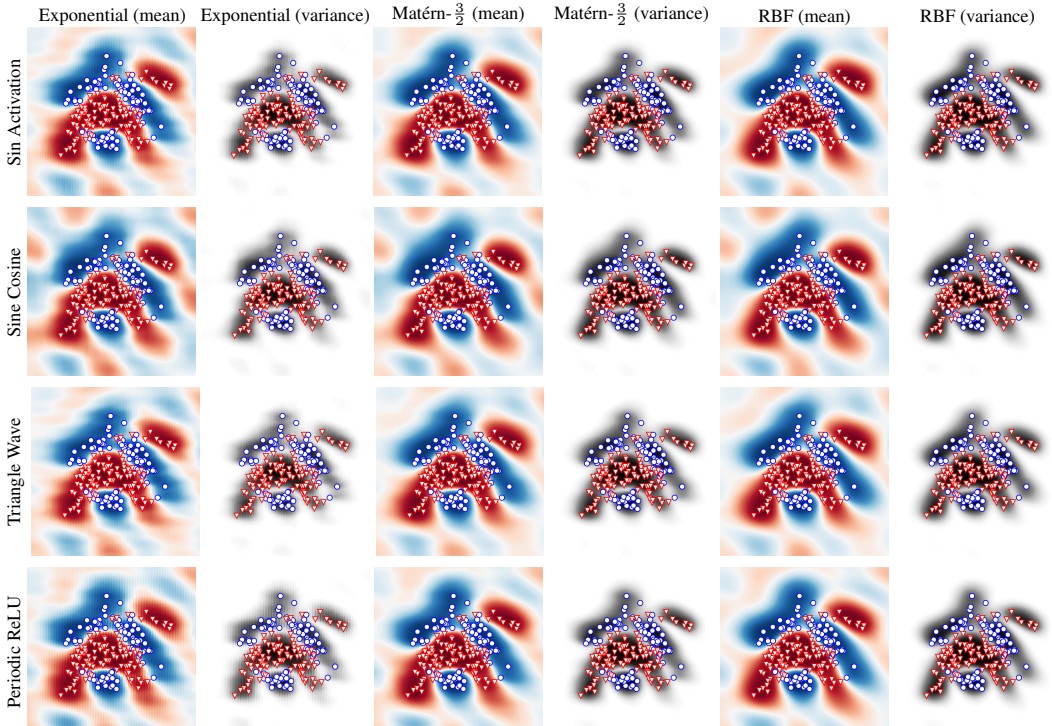

Figure 10: Posterior predictive densities of globally stationary BNNs with 30 hidden units on the banana classification task. Table rows show results for different periodic activation functions, and columns show different prior covariance functions. For each model, the resulting predictive mean and variance are plotted side by side. For predictive mean plots, the colour intensity represents confidence in the class prediction. For the variance plots, white colour represents high variance, and black colour represents low variance. We obtain comparable results regardless of the choice of the periodic activation function. Estimated using dynamic HMC run for 10k iterations and 4 chains.

The results for the boston (2–3 h), concrete (2–3 h) and airfoil (6–8 h) data sets were calculated using a single CPU per experiment, and the results for the elevators (1–2 h) data set were calculated using a single GPU per experiment. Copyright of the concrete data set: Prof. I-Cheng Yeh [62].

Table 3 shows results on four UCI regression data sets comparing deep neural networks with ReLU, locally stationary RBF [60], and locally stationary Matérn-3/2 and Matérn-5/2 [34] against global stationary models. Table 3 lists root mean square error (RMSE) and negative log predictive density (NLPD), which captures the predictive uncertainty, while the RMSE only accounts for the mean. The table lists mean and standard deviation values across folds of the 10-fold cross-validation. The values for the best performing models are shown in bold. Especially on small data sets, the standard deviation values are large, which is mostly due to differences between different folds instead of variations in model performance. The table shows that global stationary models provide better estimates of the target distribution in all cases while obtaining comparable RMSEs. Moreover, we observe that the periodic ReLU activation function tends to outperform the sinusoidal activation. Also, the importance of the choice of prior covariance can be seen in Table 3. It appears that the Matérn-3/2 covariance is the best choice for the smallest boston data set, while the smoother Matérn-5/2 or RBF covariance functions seem to be more suitable for the larger data sets.

### D.3 Benchmark Classification Tasks

The experimental setup for the UCI [8] classification tasks is the same as for the regression tasks, apart from the following details. For the UCI classification tasks, the NN architecture is a fully connected network with layers $d$-1000-1000-500-25-2000-$c$. The batch sizes used for each data set are listed in Table 4. The lengthscale parameter $\ell$ was initialized at value one.

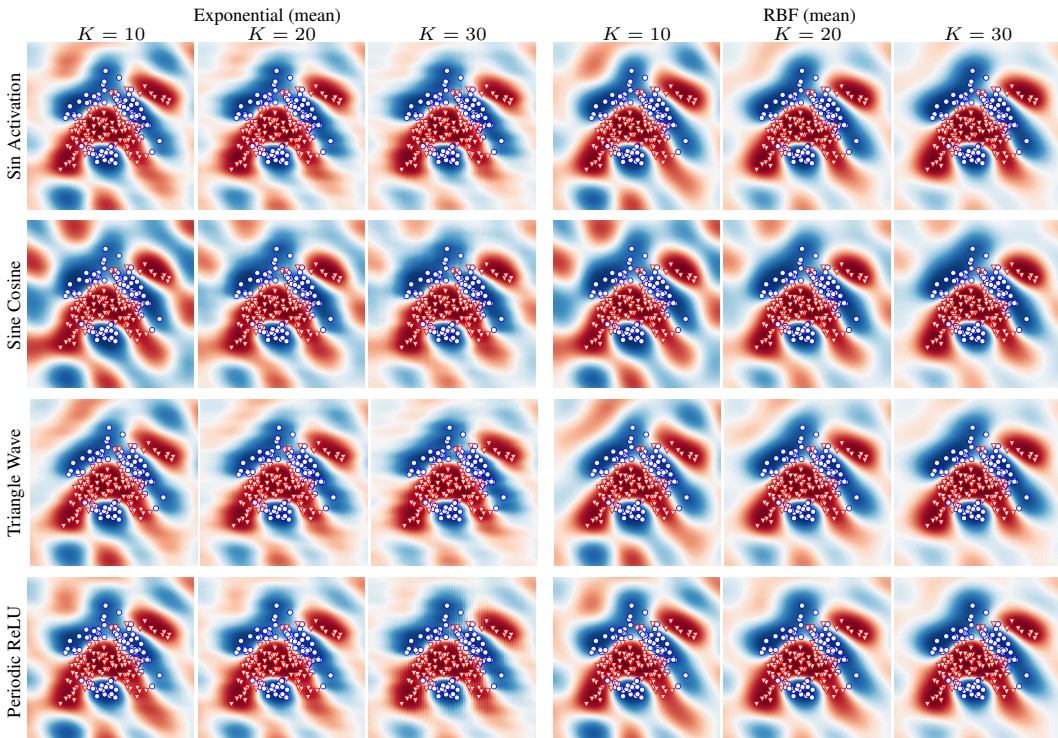

Figure 11: Posterior predictive densities of stationary BNNs for varying number of hidden units $K$ on the banana classification task. Table rows show results for different periodic activation functions, and columns show different number of hidden units, denoted as $K$. Results are shown for the two extreme cases, *i.e.* Exponential and RBF kernel. We obtain comparable results for all periodic activation function. Results are estimated using dynamic HMC run for 10k iterations and 4 chains.

Table 3: Examples of UCI regression tasks, showing the globally stationary NN model directly gives competitive mean negative log predictive density (NLPD) and root mean square error (RMSE) to locally stationary and non-stationary NN models. KFAC Laplace was used as the inference method.

| | BOSTON | | CONCRETE | | AIRFOIL | | ELEVATORS | |
| $(n, d)$ | (506, 12) | | (1030, 5) | | (1503, 5) | | (16599, 18) | |
| $(c, n_{\text{batch}})$ | (1, 50) | | (1, 50) | | (1, 50) | | (1, 500) | |
| | NLPD | RMSE | NLPD | RMSE | NLPD | RMSE | NLPD | RMSE |
|---|---|---|---|---|---|---|---|---|
| ReLU | 0.51±0.32 | 0.37±0.07 | 0.78±0.16 | 0.48±0.04 | 0.51±0.53 | 0.41±0.21 | 0.38±0.03 | 0.35±0.01 |
| loc RBF | 0.52±0.30 | 0.37±0.08 | 0.78±0.22 | 0.44±0.05 | 0.10±0.15 | 0.26±0.03 | 0.41±0.04 | 0.35±0.01 |
| glob RBF (sin) | 0.42±0.34 | 0.36±0.07 | 0.74±0.15 | 0.49±0.05 | 0.14±0.17 | 0.29±0.05 | 0.38±0.03 | 0.35±0.01 |
| glob RBF (tri) | 0.44±0.38 | 0.36±0.09 | 0.75±0.16 | 0.49±0.05 | 0.08±0.11 | 0.27±0.03 | 0.38±0.03 | 0.35±0.01 |
| loc Mat-5/2 | 0.74±0.42 | 0.36±0.07 | 0.87±0.19 | 0.47±0.04 | 0.14±0.16 | 0.27±0.03 | 0.41±0.04 | 0.35±0.01 |
| glob Mat-5/2 (sin) | 0.41±0.33 | 0.36±0.08 | 0.67±0.09 | 0.47±0.03 | 0.05±0.12 | 0.26±0.03 | 0.37±0.04 | 0.35±0.01 |
| glob Mat-5/2 (tri) | 0.45±0.38 | 0.36±0.09 | 0.65±0.09 | 0.46±0.03 | 0.05±0.16 | 0.26±0.03 | **0.37±0.03** | **0.34±0.01** |
| loc Mat-3/2 | 0.71±0.38 | 0.40±0.08 | 0.84±0.28 | **0.42±0.04** | 0.11±0.18 | 0.26±0.03 | 0.43±0.04 | 0.35±0.01 |
| glob Mat-3/2 (sin) | 0.43±0.27 | 0.39±0.08 | 0.73±0.16 | 0.49±0.05 | 0.07±0.15 | 0.27±0.03 | 0.37±0.03 | 0.35±0.01 |
| glob Mat-3/2 (tri) | 0.46±0.39 | 0.36±0.09 | 0.71±0.14 | 0.48±0.04 | 0.19±0.43 | 0.34±0.21 | 0.37±0.03 | 0.35±0.01 |
| glob RBF (sincos) | 0.50±0.36 | 0.37±0.08 | 0.73±0.13 | 0.49±0.04 | 0.19±0.21 | 0.30±0.07 | 0.38±0.02 | 0.35±0.01 |
| glob Mat-5/2 (sincos) | 0.41±0.23 | 0.39±0.08 | 0.72±0.12 | 0.49±0.04 | **0.04±0.11** | 0.26±0.03 | 0.38±0.03 | 0.35±0.01 |
| glob Mat-3/2 (sincos) | **0.35±0.32** | **0.34±0.07** | 0.68±0.14 | 0.47±0.04 | 0.05±0.16 | 0.27±0.04 | 0.56±0.54 | 0.42±0.21 |
| glob RBF (prelu) | 0.39±0.30 | 0.36±0.07 | 0.74±0.14 | 0.49±0.04 | 0.05±0.12 | **0.26±0.03** | 0.74±0.73 | 0.46±0.21 |
| glob Mat-5/2 (prelu) | 0.50±0.42 | 0.37±0.08 | **0.64±0.11** | 0.46±0.04 | 0.08±0.16 | 0.27±0.04 | 0.37±0.03 | 0.35±0.01 |
| glob Mat-3/2 (prelu) | 0.38±0.22 | 0.38±0.08 | 0.72±0.18 | 0.48±0.05 | 0.08±0.12 | 0.27±0.03 | 0.39±0.03 | 0.36±0.01 |

The results for the diabetes (3–4 h) data set were calculated using a single CPU per experiment, and the results for the adult (4–6 h), connect-4 (6–8 h) and covtype (8–12 h) data sets were calculated

Table 4: Examples of UCI classification tasks, showing the globally stationary NN model directly gives competitive accuracy and mean negative log predictive density (NLPD) to non-stationary and locally stationary NN models. KFAC Laplace was used as the inference method.

| | DIABETES (768, 8) (2, 50) | | ADULT (45222, 14) (2, 500) | | CONNECT-4 (67556, 42) (3, 500) | | COVTYPE (581912, 54) (7, 500) | |
|---|---|---|---|---|---|---|---|---|
| $(n, d)$ $(c, n_{\text{batch}})$ | NLPD | ACC | NLPD | ACC | NLPD | ACC | NLPD | ACC |
| ReLU | 0.48±0.05 | 0.76±0.04 | **0.31±0.01** | 0.85±0.00 | 0.57±0.01 | 0.82±0.00 | 0.18±0.00 | 0.93±0.00 |
| loc RBF | 0.48±0.04 | 0.76±0.04 | 0.31±0.01 | 0.85±0.00 | 0.54±0.01 | 0.81±0.00 | 0.19±0.01 | 0.93±0.00 |
| glob RBF (sin) | 0.48±0.05 | 0.77±0.04 | 0.32±0.01 | 0.85±0.00 | 0.61±0.02 | 0.81±0.00 | 0.18±0.01 | 0.93±0.00 |
| glob RBF (tri) | 0.48±0.05 | 0.77±0.04 | 0.31±0.01 | 0.85±0.00 | 0.61±0.01 | 0.82±0.00 | 0.18±0.01 | 0.93±0.00 |
| loc Mat-5/2 | 0.52±0.04 | **0.78±0.06** | 0.32±0.01 | 0.85±0.00 | **0.50±0.01** | 0.81±0.00 | 0.22±0.00 | 0.92±0.00 |
| glob Mat-5/2 (sin) | 0.48±0.04 | 0.77±0.04 | 0.31±0.01 | 0.85±0.00 | 0.64±0.03 | 0.82±0.00 | **0.17±0.01** | 0.93±0.00 |
| glob Mat-5/2 (tri) | 0.48±0.04 | 0.77±0.04 | 0.31±0.01 | 0.85±0.00 | 0.64±0.03 | 0.82±0.00 | 0.17±0.00 | **0.93±0.00** |
| loc Mat-3/2 | 0.49±0.04 | 0.76±0.05 | 0.32±0.01 | **0.85±0.01** | 0.53±0.01 | 0.81±0.00 | 0.19±0.01 | 0.93±0.00 |
| glob Mat-3/2 (sin) | 0.48±0.04 | 0.77±0.04 | 0.32±0.01 | 0.85±0.00 | 0.66±0.03 | 0.81±0.00 | 0.17±0.01 | 0.93±0.00 |
| glob Mat-3/2 (tri) | 0.48±0.04 | 0.78±0.04 | 0.32±0.01 | 0.85±0.00 | 0.66±0.02 | 0.81±0.00 | 0.18±0.01 | 0.93±0.00 |
| glob RBF (sincos) | 0.48±0.05 | 0.77±0.05 | 0.32±0.01 | 0.85±0.00 | 0.61±0.01 | 0.82±0.00 | 0.18±0.00 | 0.93±0.00 |
| glob Mat-5/2 (sincos) | 0.48±0.04 | 0.76±0.04 | 0.31±0.01 | 0.85±0.00 | 0.64±0.02 | **0.82±0.00** | 0.17±0.01 | 0.93±0.00 |
| glob Mat-3/2 (sincos) | 0.48±0.04 | 0.76±0.04 | 0.32±0.01 | 0.85±0.00 | 0.66±0.03 | 0.81±0.01 | 0.18±0.00 | 0.93±0.00 |
| glob RBF (prelu) | 0.48±0.05 | 0.77±0.04 | 0.31±0.01 | 0.85±0.00 | 0.61±0.02 | 0.82±0.00 | 0.18±0.00 | 0.93±0.00 |
| glob Mat-5/2 (prelu) | **0.47±0.04** | 0.77±0.04 | 0.32±0.01 | 0.85±0.00 | 0.64±0.02 | 0.81±0.00 | 0.18±0.00 | 0.93±0.00 |
| glob Mat-3/2 (prelu) | 0.48±0.04 | 0.77±0.05 | 0.32±0.01 | 0.85±0.00 | 0.65±0.02 | 0.81±0.00 | 0.18±0.01 | 0.93±0.00 |

Table 5: Examples of UCI classification tasks, showing the globally stationary NN model directly gives competitive area under receiver operating characteristic curve (AUC) to non-stationary and locally stationary NN models. KFAC Laplace was used as the inference method.

| | DIABETES (768, 8) (2, 50) | ADULT (45222, 14) (2, 500) | CONNECT-4 (67556, 42) (3, 500) | COVTYPE (581912, 54) (7, 500) |
|---|---|---|---|---|
| $(n, d)$ $(c, n_{\text{batch}})$ | AUC | AUC | AUC | AUC |
| ReLU | 0.84±0.03 | **0.91±0.00** | 0.90±0.00 | 0.99±0.00 |
| loc RBF | 0.84±0.03 | 0.91±0.00 | 0.90±0.00 | 0.99±0.00 |
| glob RBF (sin) | **0.84±0.03** | 0.91±0.00 | **0.90±0.00** | **0.99±0.00** |
| glob RBF (tri) | 0.84±0.03 | 0.91±0.00 | 0.90±0.00 | 0.99±0.00 |
| loc Mat-5/2 | 0.84±0.03 | 0.91±0.00 | 0.89±0.00 | 0.99±0.00 |
| glob Mat-5/2 (sin) | 0.84±0.03 | 0.91±0.00 | 0.90±0.00 | 0.99±0.00 |
| glob Mat-5/2 (tri) | 0.83±0.03 | 0.91±0.00 | 0.90±0.00 | 0.99±0.00 |
| loc Mat-3/2 | 0.84±0.03 | 0.91±0.00 | 0.90±0.00 | 0.99±0.00 |
| glob Mat-3/2 (sin) | 0.84±0.03 | 0.91±0.00 | 0.90±0.00 | 0.99±0.00 |
| glob Mat-3/2 (tri) | 0.84±0.03 | 0.91±0.00 | 0.90±0.01 | 0.99±0.00 |
| glob RBF (sincos) | 0.83±0.04 | 0.91±0.00 | 0.90±0.00 | 0.99±0.00 |
| glob Mat-5/2 (sincos) | 0.83±0.03 | 0.91±0.00 | 0.90±0.00 | 0.99±0.00 |
| glob Mat-3/2 (sincos) | 0.84±0.03 | 0.91±0.00 | 0.90±0.01 | 0.99±0.00 |
| glob RBF (prelu) | 0.84±0.03 | 0.91±0.00 | 0.90±0.00 | 0.99±0.00 |
| glob Mat-5/2 (prelu) | 0.84±0.03 | 0.91±0.00 | 0.90±0.00 | 0.99±0.00 |
| glob Mat-3/2 (prelu) | 0.84±0.03 | 0.91±0.00 | 0.90±0.01 | 0.99±0.00 |

using a single GPU per experiment. Copyright of the covtype data set: Jock A. Blackard and Colorado State University.

Table 4 and Table 5 show results on standard UCI [8] classification data sets comparing results for different activation functions. We compare a neural network with a non-stationary model using the ReLU activation function to both local stationary [34] and global stationary models for different covariance functions. Table 4 lists predictive accuracies and negative log predictive densities (NLPD) for each model. Moreover, Table 5 lists the area under the receiver operating characteristic curve (AUC) for the different models. The tables lists mean and standard deviation values across folds of the 10-fold cross-validation. The values for the best performing models are shown in bold. The classification results indicate that there are very few differences in the performance between the different models and that global stationary models achieve competitive predictive accuracy, NLPD, and AUC compared to the locally stationary and non-stationary models.

For the UCI classification tasks, we performed additional experiments using SWAG [29], inference instead of KFAC Laplace. The experiment setup here was also 10-fold cross-validation. The NN architecture is a fully connected network with layer widths d-1000-1000-500-50-c for all models. The models were trained for 20 epochs using batch sizes listed in Table 6 with Adam optimizer and a learning rate of $1 \times 10^{-4}$. The learning rate for the lengthscale parameter $\ell$ was separately set to 0.01 and initialized with one. A schedule was used for the Adam learning rates, decreasing them to one-tenth of the current value at epochs 10 and 15. The SWAG model was collected for 40 epochs (with $M = 20$ samples to estimate the covariance matrix) using SGD (momentum 0.9) as

Table 6: Examples of UCI classification tasks, showing the globally stationary NN model directly gives competitive accuracy and mean negative log predictive density (NLPD) to non-stationary and locally stationary NN models. SWAG was used as the inference method.

| | DIABETES (768, 8) (2, 50) | | ADULT (45222, 14) (2, 500) | | CONNECT-4 (67556, 42) (3, 500) | | COVTYPE (581912, 54) (7, 500) | |
|---|---|---|---|---|---|---|---|---|
| $(n, d)$ $(c, n_{\text{batch}})$ | NLPD | ACC | NLPD | ACC | NLPD | ACC | NLPD | ACC |
| ReLU | 0.53±0.07 | 0.75±0.03 | 0.38±0.15 | 0.79±0.18 | 0.54±0.12 | 0.79±0.04 | 0.19±0.00 | 0.92±0.00 |
| loc RBF | 0.49±0.05 | 0.76±0.04 | 0.33±0.01 | 0.85±0.00 | **0.47±0.01** | 0.81±0.00 | 0.19±0.00 | 0.93±0.00 |
| glob RBF (sin) | 0.51±0.05 | **0.76±0.04** | 0.35±0.03 | 0.85±0.00 | 0.51±0.05 | 0.81±0.01 | 0.18±0.00 | 0.93±0.00 |
| glob RBF (tri) | 0.53±0.07 | 0.73±0.04 | 0.33±0.01 | 0.85±0.00 | 0.51±0.05 | 0.81±0.01 | 0.19±0.01 | 0.92±0.00 |
| loc Mat-5/2 | 0.49±0.05 | 0.76±0.04 | 0.32±0.01 | **0.85±0.00** | 0.47±0.01 | **0.82±0.00** | 0.25±0.01 | 0.91±0.00 |
| glob Mat-5/2 (sin) | 0.53±0.07 | 0.74±0.04 | 0.34±0.02 | 0.85±0.01 | 0.49±0.01 | 0.81±0.00 | 0.18±0.00 | 0.93±0.00 |
| glob Mat-5/2 (tri) | 0.52±0.05 | 0.73±0.03 | 0.34±0.01 | 0.85±0.00 | 0.50±0.03 | 0.81±0.01 | 0.19±0.00 | 0.92±0.00 |
| loc Mat-3/2 | **0.49±0.04** | 0.75±0.03 | **0.32±0.01** | 0.85±0.00 | 0.47±0.01 | 0.82±0.00 | 0.23±0.01 | 0.91±0.00 |
| glob Mat-3/2 (sin) | 0.57±0.07 | 0.73±0.04 | 0.35±0.02 | 0.84±0.00 | 0.50±0.01 | 0.81±0.00 | 0.19±0.01 | 0.93±0.00 |
| glob Mat-3/2 (tri) | 0.55±0.07 | 0.74±0.04 | 0.34±0.01 | 0.85±0.01 | 0.50±0.02 | 0.80±0.00 | 0.19±0.00 | 0.93±0.00 |
| glob RBF (sincos) | — | — | 0.37±0.10 | 0.83±0.07 | 0.54±0.07 | 0.80±0.01 | 0.18±0.01 | 0.93±0.00 |
| glob Mat-5/2 (sincos) | — | — | 0.34±0.02 | 0.85±0.01 | 0.52±0.03 | 0.81±0.01 | 0.18±0.01 | **0.93±0.00** |
| glob Mat-3/2 (sincos) | 0.58±0.05 | 0.72±0.03 | 0.39±0.11 | 0.81±0.08 | 0.51±0.02 | 0.81±0.00 | **0.18±0.01** | 0.93±0.00 |
| glob RBF (prelu) | 0.52±0.06 | 0.76±0.04 | 0.34±0.01 | 0.85±0.00 | 0.50±0.01 | 0.81±0.00 | 0.19±0.01 | 0.92±0.00 |
| glob Mat-5/2 (prelu) | 0.53±0.05 | 0.75±0.03 | 0.33±0.01 | 0.85±0.00 | 0.49±0.01 | 0.81±0.00 | 0.19±0.00 | 0.92±0.00 |
| glob Mat-3/2 (prelu) | 0.58±0.07 | 0.73±0.03 | 0.34±0.01 | 0.85±0.00 | 0.49±0.01 | 0.81±0.00 | 0.19±0.01 | 0.93±0.00 |

Table 7: Examples of UCI classification tasks, showing the globally stationary NN model directly gives competitive area under receiver operating characteristic curve (AUC) to non-stationary and locally stationary NN models. SWAG was used as the inference method.

| | DIABETES (768, 8) (2, 50) | ADULT (45222, 14) (2, 500) | CONNECT-4 (67556, 42) (3, 500) | COVTYPE (581912, 54) (7, 500) |
|---|---|---|---|---|
| $(n, d)$ $(c, n_{\text{batch}})$ | AUC | AUC | AUC | AUC |
| ReLU | 0.82±0.04 | 0.87±0.12 | 0.86±0.08 | 0.99±0.00 |
| loc RBF | 0.83±0.03 | 0.91±0.00 | **0.90±0.00** | 0.99±0.00 |
| glob RBF (sin) | 0.82±0.04 | 0.90±0.01 | 0.89±0.01 | **0.99±0.00** |
| glob RBF (tri) | 0.81±0.04 | 0.91±0.00 | 0.89±0.01 | 0.99±0.00 |
| loc Mat-5/2 | 0.84±0.03 | 0.91±0.00 | 0.90±0.00 | 0.99±0.00 |
| glob Mat-5/2 (sin) | 0.80±0.05 | 0.90±0.01 | 0.89±0.01 | 0.99±0.00 |
| glob Mat-5/2 (tri) | 0.81±0.04 | 0.90±0.00 | 0.89±0.01 | 0.99±0.00 |
| loc Mat-3/2 | 0.83±0.03 | **0.91±0.00** | 0.90±0.01 | 0.99±0.00 |
| glob Mat-3/2 (sin) | 0.79±0.04 | 0.90±0.01 | 0.89±0.00 | 0.99±0.00 |
| glob Mat-3/2 (tri) | 0.81±0.04 | 0.90±0.01 | 0.89±0.01 | 0.99±0.00 |
| glob RBF (sincos) | — | 0.85±0.16 | 0.88±0.02 | 0.99±0.00 |
| glob Mat-5/2 (sincos) | — | 0.90±0.00 | 0.88±0.01 | 0.99±0.00 |
| glob Mat-3/2 (sincos) | 0.78±0.03 | 0.84±0.15 | 0.88±0.01 | 0.99±0.00 |
| glob RBF (prelu) | 0.81±0.04 | 0.91±0.01 | 0.89±0.00 | 0.99±0.00 |
| glob Mat-5/2 (prelu) | 0.81±0.03 | 0.90±0.01 | 0.89±0.01 | 0.99±0.00 |
| glob Mat-3/2 (prelu) | 0.79±0.03 | 0.91±0.01 | 0.89±0.01 | 0.99±0.00 |

the optimizer, updating the posterior estimate once per epoch. The lengthscale parameter is kept fixed during the SWAG model collection, as no SWAG posterior estimate is collected for it. The learning rate for SGD in the SWAG model collection part (the SWAG learning rate) was selected using Bayesian optimization with BoTorch in the range $(1 \times 10^{-4}, 3)$, selecting the value providing the best negative log-likelihood on the validation set. We used one-fifth of the training set of the current fold for validation, and after selecting the best SWAG learning rate we trained the model from the beginning using the full training set for each fold using the best performing SWAG learning rate. We used a fixed value for $\ell$ equal to what the earlier optimization ended at (this is to prevent ending up in a different local optimum where the optimized SWAG learning rate does not provide good results). Each models SWAG learning rate was optimized individually, but a common SWAG learning rate was used for all ten folds of a single experiment. For model averaging, 50 samples from the approximate posterior were sampled. The SWAG results for all UCI classification data sets were calculated on a single GPU per experiment and the rough running times for each data set were diabetes: 2–4 h, adult 6–8 h, connect-4 8–10 h and covtype 15–20 h. For the SWAG results, Table 6 lists predictive accuracies and NLPDs, and Table 7 lists AUC values for the different models. The results show that similar to the KFAC Laplace results, the global stationary models using periodic activation functions achieve competitive predictive accuracy, NLPD, and AUC compared to the locally stationary and non-stationary models. The main difference to results obtained using KFAC Laplace is that SWAG seems to produce more variability in the results between different models. The missing values in Table 6 and Table 7 are due to the optimization diverging in the SWAG model collection phase.

## D.4    Detection of Distribution Shift with Rotated MNIST

For the MNIST ([25], available under CC BY-SA 3.0) digit classification experiment, the feature extractor part of the NN architecture has two convolutional layers (32 and 64 channels, both using a $3 \times 3$ kernel) followed by a fully connected layer taking the dimensionality down to 25, and the following model layer has 2000 hidden units. The models were trained on the MNIST training set for 50 epochs using a batch size of 64 with an SGD optimizer (learning rate $1 \times 10^{-3}$, momentum $0.9$), using only unrotated images. A schedule was used for the SGD learning rates, decreasing them to $0.9$ of the current value at epochs 25 and 37. The learning rate for the lengthscale parameter $\ell$ was set to $1 \times 10^{-4}$, and was initialized with $0.2$. The posterior inference was performed using KFAC Laplace [45] with a fixed variance scale of one. For model averaging, we used 30 posterior samples. We tested the trained model on the standard unrotated MNIST test set and rotated versions of the same test set for rotation angles every $10°$ up to $360°$. Running this experiment for one model on one GPU took roughly 2 hours due to testing the model on multiple test sets.

Fig. 5 shows the results on the rotated MNIST experiment for different models. We evaluate the predictive accuracy, mean confidence of the predicted class, and NLPD on the rotated test sets. The results indicate that all models obtain similar accuracy results, while only local and global stationary models do not result in over-confident uncertainty estimates. For an ideally calibrated model, the mean confidence would decrease as low as the accuracy curve when the digits are rotated, which would keep the NLPD values as low as possible. We see that even for the local and global stationary models, the mean confidence has a minimum of around $0.55$ while the accuracy decreases below $0.2$. We also observe that the NLPD curves rise to high values (over 3) for the better performing models. The accuracy of all models increases near the $180°$ rotation. This is most likely due to numbers 0, 1 and 8 appearing similar with $0°$ and $180°$ rotations. Interestingly, the NLPD values for the local and global stationary models hardly decrease for the rotation angle of $180°$ although the accuracy for this angle increases compared to adjacent angles. This could be due to number 6 looking like number 9 at $180°$ rotation, and vice versa, causing the model to make overconfident incorrect predictions. Although, this kind of overconfident misclassification cannot be prevented even with a correctly calibrated model, as samples of one class genuinely appear to belong to another class.

Fig. 12 shows additional results on the rotated MNIST experiment for different models, using only a maximum a posteriori (MAP) estimate for the model parameters. The models used for the MAP results are the same trained models that were used for the results in Fig. 5, but for the MAP results the KFAC Laplace inference step is skipped. The MAP results are almost identical to the KFAC Laplace results, except that $90°$ and $270°$ rotations for mean confidence have slightly higher values for the MAP results. This suggests that the KFAC Laplace inference might not be very successful in improving data set shift detection properties in this experiment.

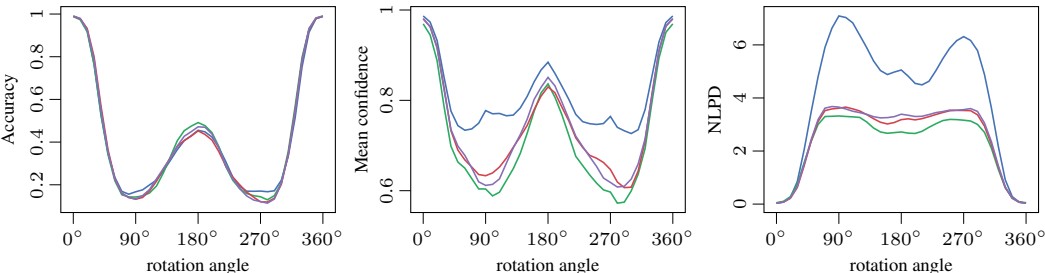

Figure 12: Rotated MNIST maximum a posteriori (MAP) results: The models have been trained on unrotated digits. The test-set digits are rotated at test time to show the sensitivity of the trained model to perturbations. Model predictions are based on a MAP estimate of model parameters. All models perform equally in terms of accuracy, while ReLU (——) shows overconfidence in terms of mean confidence and NLPD. The stationary RBF models (—— local, —— sin, —— sin–cos) capture uncertainty.

## D.5    Out-of-distribution Detection Using CIFAR-10, CIFAR-100, and SVHN

For this experiment, the feature extractor part is a GoogLeNet [52] followed by a 512 node wide model layer. Pre-trained weights are used for the feature extractor part of the NN, and kept unchanged during the model training (pre-trained model from `https://github.com/huyvnphan/PyTorch_CIFAR10`). The models were trained on CIFAR-10 [23] for 20 epochs using a batch size of 128 with Adam optimiser and a learning rate of $1 \times 10^{-4}$. The learning rate for the lengthscale parameter $\ell$ was separately set to $1 \times 10^{-5}$, and the lengthscale parameter was initialised with one. The posterior inference was performed using SWAG [29]. The SWAG model was collected for 40 epochs ($M = 20$) using SGD (momentum 0.9) as the optimiser, updating the posterior estimate once per epoch. The lengthscale parameter is kept fixed during the SWAG model collection. The learning rate for SGD in the SWAG model collection part (the SWAG learning rate) was selected using Bayesian optimisation with BoTorch in the range $(1 \times 10^{-4}, 3)$ based on the negative log-likelihood on the validation set. As validation data, we used the CIFAR-10 test set. Using the CIFAR-10 test set as the validation set for selecting hyperparameters is valid here; as for this experiment, the focus is not on measuring the performance on the test set but evaluating OOD detection performance on the CIFAR-100 and SVHN test sets. After selecting the best SWAG learning rate, we trained the model from the beginning using the best performing SWAG learning rate and a fixed value for $\ell$ equal to the earlier optimisation. For each model, the SWAG learning rate was optimised individually. For model averaging, 30 samples from the approximate posterior of the parameters were sampled to calculate the predictions on CIFAR-10, CIFAR-100 [23], and SVHN [39]. Running this experiment for one model on one GPU took roughly one day due to the Bayesian optimization process.

Fig. 14 compares model performance on out-of-distribution detection for image classification for non-stationary, local stationary and global stationary models. Both CIFAR-100 (more similar) and SVHN (more dissimilar to CIFAR-10) images are OOD data, and the models should show high uncertainties (high predictive entropy, high predictive variance) for the respective test images. The histograms of predictive entropies for different test sets show that most models can separate between in-distribution and OOD data based on this metric. However, the predictive marginal variance histograms show that the global stationary models can better detect the OOD samples compared to ReLU and local stationary models. Interestingly, the ReLU model shows higher variance on CIFAR-100 images compared to SVHN images, although SVHN images are more different from the training set images. For global stationary models, both entropy and variance histograms show that the models clearly consider SVHN more OOD than CIFAR-100, which is intuitive as CIFAR-100 resembles CIFAR-10 more than SVHN. Fig. 14 also shows sample images for most/least similar to the training data distribution that the model has learned. Looking at these images for different models, we can see that for CIFAR-10, images with dark background result in high uncertainty for all models. For the CIFAR-100 sample images, the global stationary Matérn-$3/2$ model has classified pictures of animals and humans with the highest confidence, which seems intuitive as these could be considered resembling some of the CIFAR-10 classes (for example, dogs or cats). Moreover, the images with the highest uncertainty seem visually very different from CIFAR-10 images. For the SVHN sample images, all models seem to be most confident about clear and sharp images and blurry images result in high uncertainty, which is reasonable as CIFAR-10 images usually have clear shapes. However, this is again most apparent for the global stationary models, suggesting the model has learned meaningful representations of the input space.

Using the same results that are visualized in the histograms in Fig. 14, we calculated area under receiver operating characteristic curve (AUC) and area under precision-recall curve (AUPR) values for OOD detection in the CIFAR-10 experiment to provide additional quantitative results, treating either CIFAR-100 or SVHN as the OOD data set. We calculated the AUC and AUPR measures using the marginal variance as the metric to determine whether a sample is OOD or not. We consider this a better metric for OOD detection compared to predictive entropy, as in-distribution samples that are hard to classify are expected to have high predictive entropy, not necessarily allowing the detection of OOD samples based on this metric. Table 8 lists the calculated AUC and AUPR numbers. The results for SVHN as the OOD data set indicate a clear difference between the non-stationary (ReLU) and the stationary (local and global) models. The globally stationary RBF model achieves the best AUC score. Treating CIFAR-100 as an OOD set is not as straightforward considering numerical AUC and AUPR comparisons. CIFAR-100 images are visually very similar to CIFAR-100 images but representing different classes, and hence can be considered not strictly OOD. For example, it is reasonable to expect that even a correctly operating model may consider some CIFAR-100 images

Table 8: Table of numerical results on the image classification OOD task. The results used to calculate the numbers in this table are the same that were used to create histograms in Fig. 14. The table lists the area under receiver operating characteristic curve (AUC) and the area under precision-recall curve (AUPR) for each of the models. Numbers are calculated both for considering CIFAR-100 or SVHN as the OOD set, while CIFAR-10 is the in-distribution data. The table also visualizes whether each model considers CIFAR-100 or SVHN more OOD based on which data set is detected as OOD more effectively.

| | AUC | | | AUPR | | |
|---|---|---|---|---|---|---|
| OOD data set | CIFAR-100 | | SVHN | CIFAR-100 | | SVHN |
| ReLU | 0.974 | > | 0.961 | 0.963 | < | 0.970 |
| loc RBF | **0.976** | < | 0.987 | **0.976** | < | **0.995** |
| glob RBF (sin) | 0.942 | < | **0.988** | 0.940 | < | **0.995** |
| loc Mat-3/2 | 0.973 | < | 0.983 | 0.972 | < | 0.993 |
| glob Mat-3/2 (sin) | 0.965 | < | 0.981 | 0.965 | < | 0.993 |

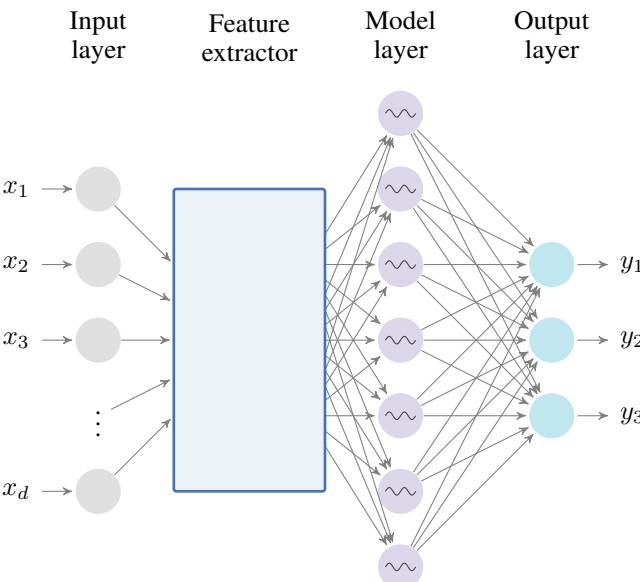

Figure 13: An illustrative figure describing the model architecture. The model first passes an input $\mathbf{x} \in \mathbb{R}^d$ into a feature extractor. The feature extractor part is a task dependent neural network architecture, which can be for example a fully connected structure or some convolutional layers. The feature extractor results in some $L$-dimensional representation (in the illustration $L = 3$), which is followed by a fully connected hidden layer (referred to as model layer in the text) resulting in a $K$ dimensional representation ($K = 7$ in the figure). The model specific activation function is applied on this $K$-dimensional representation (sinusoidal activation in the figure). The output $\mathbf{y} \in \mathbb{R}^c$ is produced by a fully connected output layer. Here $c$ is the number of classes in case of a classification task ($c = 3$ in the figure).

more in-distribution than the most difficult or visually peculiar CIFAR-10 test images. For this reason, it is reasonable to compare AUPR and AUC numbers of each model for the two OOD data sets, CIFAR-100 and SVHN, and observe which data set is considered more OOD. We expect SVHN to be considered more OOD, which is true except for the ReLU model based on the AUC metric.

## Results with ReLU model

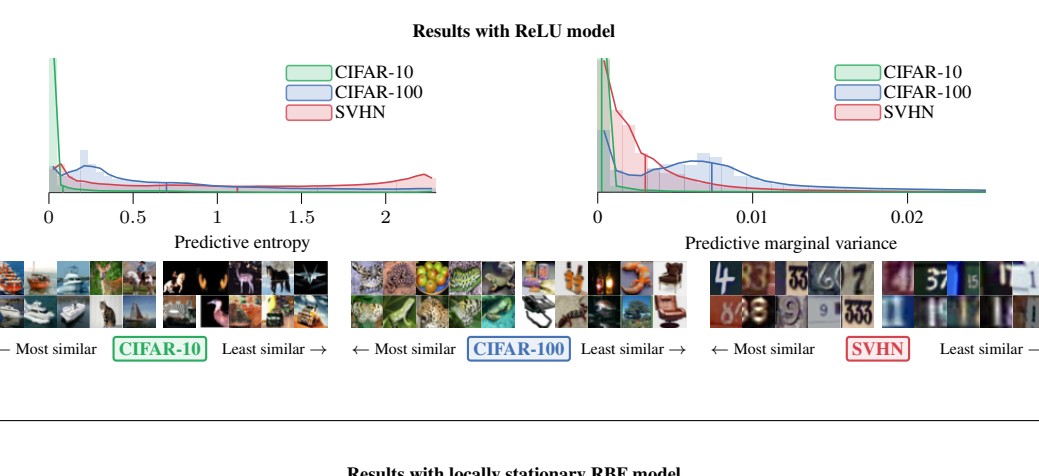

## Results with locally stationary RBF model

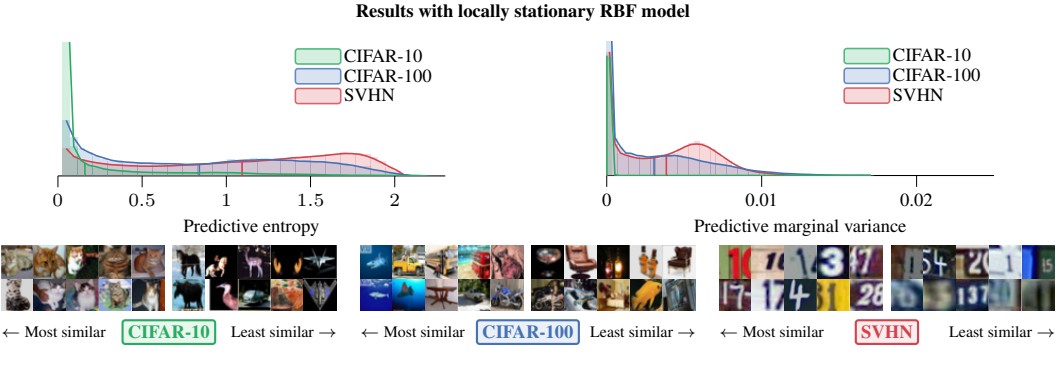

## Results with globally stationary RBF model (sinusoidal)

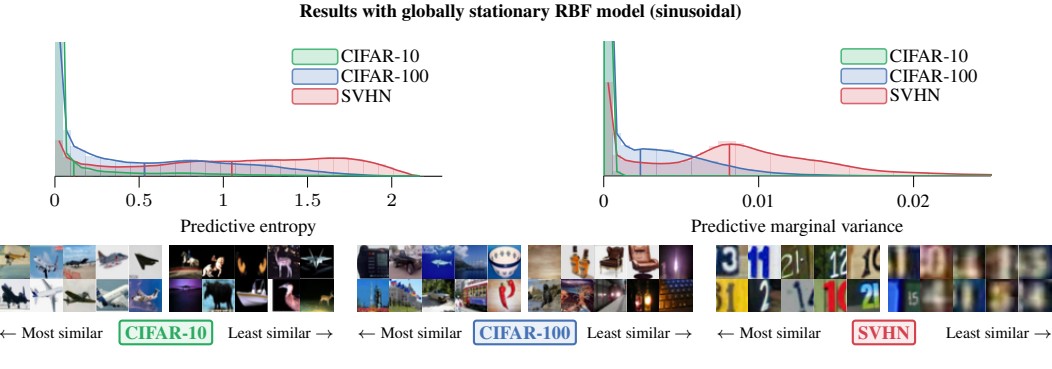

## Results with locally stationary Matérn-$\frac{3}{2}$ model

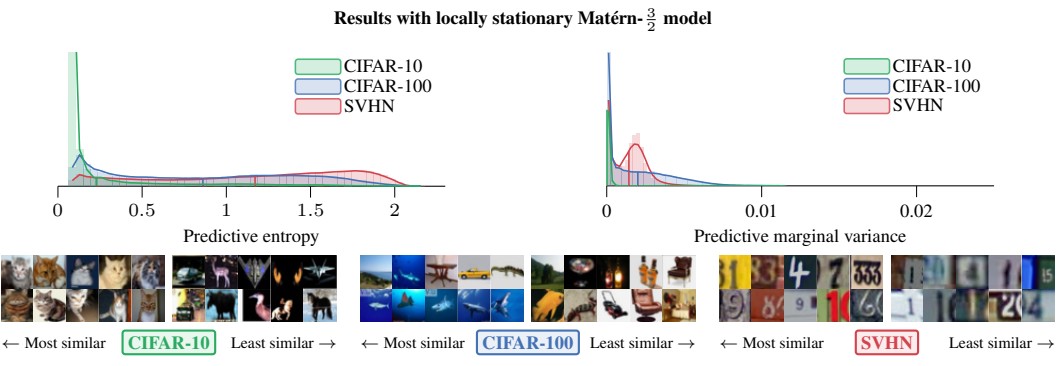

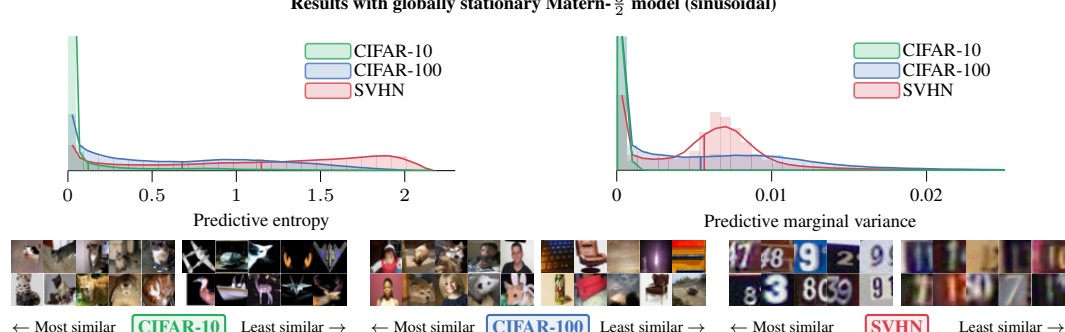

Figure 14: OOD detection experiment results for models trained on CIFAR-10 and tested on CIFAR-10, CIFAR-100, and SVHN. Predictive entropy histograms of test image results are on the left, and predictive marginal variance histograms are on the right. On the bottom are sample images from each test set: left-side images with lowest entropy/highest confidence, and right-side images with highest entropy/lowest confidence.