# OpenReview forum: "Periodic Activation Functions Induce Stationarity"
_NeurIPS.cc/2021/Conference — NeurIPS 2021 Poster_

### Official Review · Reviewer_8dcJ · 2021-07-13

**Rating:** 7
**Confidence:** 4

**Summary:**

The authors show that under the infinitely wide Bayesian neural network framework, periodic activation functions combined with student-t distributed readout parameters result in a GP with the Matern class of covariance functions. Periodic activations are considered in a relatively wide sense, including not only cosine and sine but also triangle and deep-learning inspired "periodic ReLU". The Matern class, being a stationary covariance function with intuitive and controllable smoothness properties, sees application in a diverse range of tasks. The authors compare predictive uncertainties and performance of their new infinite-width and finite-width models against corresponding locally stationary and non-stationary models on some toy and standard benchmarks.

**Limitations And Societal Impact:**

N/A.

**Main Review:**

Strengths:
a. The paper addresses an important gap in research. More specifically, it is known (Meronen et al.) that a certain activation function induces a locally stationary covariance function corresponding with an appropriately decayed Matern kernel. However, it is not known how to construct the globally stationary Matern class. This paper addresses this problem.
b. The paper is very well written and easy to read. I did not spot any typos or awkward sentences. The paper uses elegant tools that are accessible to most readers in the community.
c. The experiments and illustrations are insightful.

Limitations/suggestions for improvement:
1. I am not convinced that stationarity of the infinite-width model transfers to stationarity of the finite-width model when the width is small. More importantly, it is not necessarily true that the mean-reverting property holds for finite-width models. Ideally the authors would provide some quantification of error of this fact in the finite-width setting.
2. The authors do not mention the related infinite-width neural tangent kernel (NTK) models. While not Bayesian, these are also GPs in the infinite width and do offer what some would call an uncertainty. The form of the covariance function is very similar to the Bayesian setting. Do your stationarity results carry over to this setting? If not, what breaks or presents a technical difficulty? At the least, I think the authors should mention this work and describe how it relates to their results.
3. There are some minor issues concerning the rigour of the proofs in A.3 and A.4. I believe the result is correct, but the argument can be tightened. In (32), the authors approximate the wave by a truncated series. In (35) the put this truncation inside the kernel. This is the truncated series, not the full infinite series. There should be a limit inside the integral which needs to be interchanged with the integral via dominated convergence argument or similar.  There is another issue around (45) where are limit should be present and then the \approx may be removed. The same criticism applies to A.4.

Disclaimer:
I did not review the code provided in the supplementary material. I did not read appendix D.


**Time Spent Reviewing:**

2

---

> ### Author Response · Authors · 2021-08-09
> **Response to 8dcj**
>
> We thank the reviewer for their enthusiasm and detailed comments. We address the comments and concerns below in the order received.
>
> ### Comment 1
>
> > I am not convinced that stationarity of the infinite-width model transfers to stationarity of the finite-width model when the width is small. More importantly, it is not necessarily true that the mean-reverting property holds for finite-width models. Ideally the authors would provide some quantification of error of this fact in the finite-width setting.
>
> Our experimental results suggest that stationarity properties transfer to finite-width models surprisingly well, even when the width is small. Our toy experiments are performed with 30 hidden units, and the UCI classification and regression tasks use 50 hidden units in the model layer. The toy experiment results show good correspondence to the infinite-width GP models; see Figures 7 and 8 in the appendix. We will add additional simulation results with varying numbers of hidden units to the appendix as well as quantitative results based on the estimated Gram matrices.
>
> ### Comment 2
>
> > The authors do not mention the related infinite-width neural tangent kernel (NTK) models. [...] Do your stationarity results carry over to this setting? If not, what breaks or presents a technical difficulty? At the least, I think the authors should mention this work and describe how it relates to their results.
>
> We agree that adding a discussion on the NTK into the related work is a good idea. Whether the stationarity results carry over to the NTK is indeed an interesting one. As the NTK is defined in terms of the derivative of the neural network function w.r.t. the parameters, we expect our results carry over in the case of sinusoidal activations. We believe a detailed analysis of the evolution of neural networks with periodic activation functions during gradient descent to be potentially very insightful but will leave it open for future work.
>
> ### Comment 3
>
> > There should be a limit inside the integral which needs to be interchanged with the integral via dominated convergence argument or similar.
>
> We thank the reviewer for the thorough review of our derivations and for pointing out that the derivations can be improved in rigour. We will revise the derivations accordingly to strengthen the argument by considering the limit of the series instead of a truncated series as suggested.

---

### Official Review · Reviewer_qVhs · 2021-07-15

**Rating:** 7
**Confidence:** 3

**Summary:**

The paper derives a class of Gaussian processes, derived as infinite-width limits of neural networks, that exhibit stationary covariance functions. The key observation comes from a comparison between the expression of a stationary kernel given by Bochner's theorem, and the infinite-width kernel of a neural network. Stationary covariance functions can be guaranteed whenever the activation function is chosen to be periodic. Numerical experiments suggest that these GPs enjoy improved performance on regression tasks, and more accurate uncertainty prediction on out-of-distribution inputs.

**Limitations And Societal Impact:**

One common issue with Bayesian neural networks lies in its occasional overconfidence. This paper aims to address that issue with a simple trick. There are always some risks associated with uncertainty prediction, but this comes with the territory. The authors have pointed out a few limitations of their methodology in the conclusions; I believe this suffices.

**Main Review:**

I very much enjoyed reading this paper. The authors present a simple thought-provoking observation with sufficient mathematical justification and an adequate selection of experiments to highlight the advantages. The paper is well-written, with excellent visual illustrations throughout to demonstrate the authors' claims. I found the literature review fairly comprehensive. Overall, I believe the authors have ticked all the boxes for a great conference paper.

A few comments:

- I would suggest being more careful about claiming improved NLPD in Table 1. The error margins due to the CV are large, and the 'improvements' shown in mean don't seem to be sufficient to suggest a significant difference.

- It's unfortunate that Figure 6 does not contain a comparison to a locally stationary model; this comparison being relegated to the supplementary material. I understand this was done for space constraints, but I feel a better compromise could be made. Maybe by restricting to predictive marginal variance in the main text, and relegating entropy to supplementary material?

- 'Banana' (dataset) is in typewriter format earlier in the paper, but not in the experiments.

**Time Spent Reviewing:**

3

---

> ### Author Response · Authors · 2021-08-09
> **Response to qVhs**
>
> First, we want to thank the reviewer for the enthusiasm about our work and the detailed comments. We will address the comments and concerns in the order received.
>
> ### Comment 1
>
> > I would suggest being more careful about claiming improved NLPD in Table 1. The error margins due to the CV are large, and the 'improvements' shown in mean don't seem to be sufficient to suggest a significant difference.
>
> The large standard deviations in Table 1 are due to the small number of observations and the fact that some splits in the 10-fold CV end up being significantly harder than others. We will improve the wording of the paper and point out the variability introduced by differences between the random splits.
>
> ### Comment 2
>
> > It's unfortunate that Figure 6 does not contain a comparison to a locally stationary model; this comparison being relegated to the supplementary material. I understand this was done for space constraints, but I feel a better compromise could be made.
>
> This was indeed a choice made due to space constraints. We will use the additional page in the camera-ready version to include the globally and locally stationary results in Figure 6 (main text), which are currently split into Figure 6 and Figure 9 in the appendix.
>
> ### Comment 3
>
> > 'Banana' (data set) is in typewriter format earlier in the paper, but not in the experiments.
>
> We will correct this typo in the camera-ready version.

---

### Official Review · Reviewer_C4fJ · 2021-07-16

**Rating:** 6
**Confidence:** 4

**Summary:**

The authors attempt to draw a connection between
(1) the spectrum $S(\omega)$ of stationary kernels, e..g K(x, x') = k(x-x') and
(2) the prior $p(\omega)$ of the weights in random initialized networks of the form $\psi(wx +b)$, where the activation function $\psi$ is periodic
by using the well-known Bochner’s theorem.

**Limitations And Societal Impact:**

Limitations of the work hasn't been brought up in the paper;
One side note:
There has been significant advances in infinite-width networks that dramatically generalize Neal's work, which hasn't been discussed / cited in the paper. See e.g. "Deep Neural Networks as Gaussian Processes", "Gaussian Process Behaviour in Wide Deep Neural Networks" and reference therein.


**Main Review:**

First, I summarize (simplify) the arguments of the paper and points out a couple mistakes of the paper. The main result of the paper is trying to say that the feature (or NNGP) kernel of a randomly initialized one-hidden layer network with weight prior $p(w)$ can be approximated written as (up to some scalars, positive/negative signs)
\begin{align}
 K(x, x') \approx  C_{\psi} \int p(w) e^{i\omega^T (x -x')} d\omega
\end{align}
This approximation is not very correct in general. Even it is, the formula itself is not significant.

Here is the main argument to "get" the above formula.
\begin{align}
K(x, x') = \int _\omega p(w) \int_b \psi(wx+b) \psi(wx'+b) p(b) db dw
= \int _\omega p(w)  \psi \ast \bar \psi (\omega (x-x'))  dw
\end{align}
where the following equation used the fact that $b$ is uniformly supported on $[-\pi, \pi]$ and $\psi$ is a periodic function on that and thus the inner integral is a convolution on the torus
\begin{align}
\int_b \psi(wx+b) \psi(wx'+b) db  = \psi \ast \bar \psi (\omega x-  \omega x'))
\end{align}
Here $\bar \psi( x) = \psi(-x) $.  We can then apply Fourier expansion to $\psi \ast \bar \psi $ since it is periodic and write
\begin{align}
\psi \ast \bar \psi (z) =  \sum  c_k  \bar c_k e^{ik\pi z }
\end{align}
Plugging back the original equation, one has
\begin{align}
 K(x, x') =  \int p(\omega) \sum c_k \bar c_k e^{ik\pi \omega(x-x') }  d\omega
\end{align}
This clearly implies $K$ is stationary. Here is what goes wrong in the paper. The authors made the claim that
\begin{align}
 K(x, x') =  \int p(\omega) \sum c_k \bar c_k e^{ik\pi \omega(x-x') }  d\omega  \sim   \int p(\omega) c_1 \bar c_1 e^{i\pi \omega(x-x') }  d\omega
\end{align}
That is keeping $k=1$ frequency and getting rid of all others! This is OK for $\cos/\sin$ combination seems $c_k=0$ for $k\neq1$. For others,  the case $k=0$ is ok but need to assume $\psi$ has mean zero (I haven't seen this assumption in the paper while all examples in the papers are odd functions.) but NOT OK for other small $k$ (e.g. k=2, 3 etc.) In equation (40) of the appendix, the authors conclude $c_k$ can be dropped due to rapid decay, which is clearly WRONG! Even though the tail terms could be dropped, but the lower frequency terms shouldn't.

Empirically, applying periodic activation functions to OOD problem seems interesting, but this is idea seems not original; see e.g. "Stationary Activations for Uncertainty Calibration in Deep Learning".

Overall, the originality, quality and significancy of the paper is not very high and I don't recommend acceptance of it.


**Time Spent Reviewing:**

4

---

> ### Author Response · Authors · 2021-08-09
> **Response to C4fJ**
>
> First, we would like to thank reviewer C4fJ for thoroughly reviewing the derivations for the triangular wave and the periodic ReLU in the appendix.
>
> ### Comment 1
> > The main result of the paper is trying to say that the feature (or NNGP) kernel of a randomly initialized one-hidden layer network with weight prior p(w) can be approximated written as [...]
>
> As noted by the other reviewers and stated in our submission, the main contribution of our work is to show that the use of periodic activation functions in neural networks induces global stationarity and establishes a connection between the prior distribution on the weights of the random neural network and the spectral density of the stationary limiting Gaussian process. Thus, periodic activation functions induce a strong inductive bias (stationarity) and allow the practitioner to encode modelling assumptions concisely by exploiting the connection between the prior on the weights and the spectral density of the limiting process. This has been illustrated for the Matérn class of kernel functions, containing some of the most frequently used kernels in the Gaussian process literature.
>
> ### Comment 2
> > In equation (40) of the appendix, the authors conclude ck can be dropped due to rapid decay, which is clearly WRONG!
>
> We agree that omitting terms with $k > 0$ can be problematic, but would like to point out that the purpose of the approximation is merely to provide an approximate link to the spectral density of the Matérn class for the triangular wave and the periodic ReLU. Furthermore, derivations in question establish the connection between the prior for the weights and the spectral density even without the approximation proposed in the appendix.
> We will now provide an alternative perspective on the derivations. Instead of truncating the Fourier series, it is also possible to write the density on the weights in Eq. (40) in the form of a mixture density (assuming the density $p$ is in the location-scale family, which is the case for the prior distributions discussed in the paper):
>
> $$
> \\kappa(x,x') = \\int \\sum^{n-1}_{k=0} \\frac{(-1)^{2k}}{\\lambda^4_k} p(w) e^{(\\mathrm{i} \\lambda_k w (x - x'))}
> \\mathrm{d}w
> $$
>
> $$
> \\kappa(x,x') = \\int \\sum^{n-1}_{k=0} \\pi_k p(w) e^{(\\mathrm{i} \\lambda_k w (x - x'))} \\mathrm{d}w
> $$
>
> $$
> \\kappa(x,x') = \int \sum^{n-1}_{k=0} \pi_k p(w | \lambda_k) e^{(\mathrm{i}  w (x - x'))} \mathrm{d}w
> $$
>
> , where $p(w | \lambda_k)$ denotes the density function of $p(w)$ with scale parameter $\lambda_k$. Let us now denote $\sum^{n-1}_{k=0} \pi_k p(w | \lambda_k)$ as $\hat{p}(w) $, i.e.,
>
> $$
> \\kappa(x,x') = \int \hat{p}(w) e^{(\mathrm{i}  w (x - x'))} \mathrm{d}w \\, .
> $$
> Finally, by letting $r = x − x'$ , we find that we again recover the spectral density decomposition of a stationary process given by the Wiener–Khinchin theorem.
>
> We again obtain a connection between the prior (in this case, a mixture) and the spectral density through the Wiener–Khinchin theorem. In this case, the spectral density also has to admit a mixture density. However, working with a mixture density as prior could be potentially challenging for inference. Fortunately, the error introduced when approximating the mixture through its first component is neglectable. To verify this claim, we compared the exact kernel functions (Exponential, Matern-3/2 and RBF) against simulations for each of the four different activation functions. The results are shown in Figure 4 (main text). All four periodic activation functions recover the behaviour of the three kernel functions, which would not happen if the approximation proposed in the appendix would entail strong errors. Additionally, the empirical results in the paper and the appendix (see Figure 7, Figure 8 and Table 3–7) all provide strong evidence that the approximation does not entail strong errors as results would otherwise not be comparable across activation functions. In addition to this strong evidence and the simulation results shown in the main text, we will provide additional simulation results containing an additional quantitative assessment of the introduced error in the appendix of the camera-ready paper.
> We will also provide a detailed discussion on the approximation and the introduced error in the appendix of the camera-ready version.
>
> ### Comment 3
> > [...] applying periodic activation functions to OOD problems seems interesting, but this is idea seems not original; see e.g. "Stationary Activations for Uncertainty Calibration in Deep Learning"
>
> The work by Meronen et al. [31] only considers local stationarity and does not utilise or discuss periodic activation functions. In contrast, our work shows that periodic activations functions induce global stationarity and establish a connection between the prior on the weights and the spectral density of the limiting Gaussian process.
>
> ### Comment 4
> > Overall, the originality, quality and significancy of the paper is not very high and I don't recommend acceptance of it.
>
> We politely disagree and would like to cite the comments by the other reviewers:
>
> > Drawing an explicit parallel between the weight distribution in a neural network and the spectral density of the covariance function of its limiting GP is a strong theoretical result. (6Fuy)
>
> > The main theoretical result in this paper is a significant step towards a better understanding of the link between neural architecture, weight distribution and inductive bias. (6Fuy)
>
> > I believe the authors have ticked all the boxes for a great conference paper. (qVhs)
>
> > The paper addresses an important gap in research. (8dcj)
>
> ### References
> [31] L. Meronen, C. Irwanto, and A. Solin. Stationary activations for uncertainty calibration in deep learning. In Proceedings of Advances in Neural Information Processing Systems (NeurIPS), pages 2338–2350. Curran Associates, Inc., 2020.

---

> > ### Comment · Reviewer_C4fJ · 2021-08-26
> > **update**
> >
> > Thanks for the detailed reply.
> >
> > I am not comfortable to raise the score to acceptance unless the authors can (or promise to) resolve the following two questions:
> >
> > (1.) Error bound for removing all  $\geq k$ frequencies.
> >
> > (2.) The required prior $p(w)$ is heavy-tailed in general. An error control in terms of $n$, the number of required sampling, aka, the width. This is very relevant given the prior is not sub-Gaussian or even sub-exponential.

---

> > > ### Author Response · Authors · 2021-08-27
> > > **Re: update**
> > >
> > > Thank you for getting back to us. Regarding (1), we agree that it is a good idea to address the effect of removing $\geq k$ frequencies in the derivation of the triangular and periodic ReLU activations. We will aim to quantify these effects in App. A.3 and A.4.
> > >
> > > Regarding (2), we are happy to add discussion on the effect of the finite width model compared to the infinitely wide GP prior. As you note, these effects become more pronounced with heavy-tailed priors, and similar analysis appears in rank-reduced GP literature for low-order Matérn kernels. As mentioned in our earlier reply to Reviewer 8dcj, we will also be adding an empirical study of the effect of varying the number of hidden units.

---

> > > > ### Comment · Reviewer_C4fJ · 2021-08-27
> > > > **update**
> > > >
> > > > Thanks! I update the score accordingly.

---

### Official Review · Reviewer_6Fuy · 2021-07-16

**Rating:** 7
**Confidence:** 3

**Summary:**

The author shows that periodic activation functions in a single layer Bayesian neural networks induce global stationarity in the limiting Gaussian Process. They show a direct link between the weight distribution (before training) and the spectral density of the covariance limiting GP, for various periodic activation functions. They leverage this observation to build models that are more robust to out-of-domain data.

**Main Review:**

I enjoyed reading the paper: I found it clearly written and with appropriate background for a non-expert audience. This work is timely and, I think relevant to the sub-community interested in coordinated-based networks/"implicit representations" as those bloom in many different applications (NerF, occupancy networks, SDF shape representation, GANs etc.).

The author demonstrates an equality between the prior weight distribution in a single layer neural network and the spectral density of the stationary covariance function of the limiting Gaussian Process, for four types of periodic activation functions. Thanks to this equality, the author shows that taking a Student-t distribution over the weights induces a Matern covariance function for the limiting GP. They also demonstrate that BNN with stationary limiting GP are less vulnerable to out-of-domain data.

Drawing an explicit parallel between the weight distribution in a neural network and the spectral density of the covariance function of its limiting GP is a strong theoretical result. As the author shows, one can deduce a weight distribution from the stationary covariance kernel they want to induce, for any fully-connected neural network having periodic activation functions.

The author do not use the proven parallel between Student-t distribution and Matern covariance in their experiments. However, they experimentally show the advantage of stationarity in Bayesian neural network by proving robustness to our-of-domain data for different regression and classification task.

The main theoretical result in this paper is a significant step towards a better understanding of the link between neural architecture, weight distribution and inductive bias. The scope of the experiments described in this paper is limited to proving the advantage of a stationary limiting GP.

**Time Spent Reviewing:**

2

---

> ### Author Response · Authors · 2021-08-09
> **Response to 6Fuy**
>
> First, we want to thank the reviewer for the constructive comments and the excitement about our work. We will address the comments and concerns in the order received.
>
> ### Comment 1
> > The authors do not use the proven parallel between Student-t distribution and Matern covariance in their experiments.
>
> The derived parallel between the Student-t distribution and Matérn covariance functions was indeed leveraged in the experiments. We conducted all experiments with all of the presented periodic activation functions and discussed priors, i.e. Cauchy, Student-t, and Gaussian. However, due to space constraints, we restricted the presented results in the main paper to a subset of these. The results for all periodic activation functions and each of the discussed priors are included in the appendix. The priors affect the initialisation of NN weights and the loss function during the NN training as we maximise the log joint. We will improve the description of the experiments to make this clear in the revised version.
>
> ### Comment 2
> > The scope of the experiments described in this paper is limited to proving the advantage of a stationary limiting GP.
>
> In addition to the empirical evaluation of the performance of stationary inducing Bayesian neural networks, we provide simulation results (see Fig. 4) to verify the correctness of our derivations.

---

### Decision · Program_Chairs · 2021-09-28

**Decision:**

Accept (Poster)

**Comment:**

This paper shows that periodic activations in a Bayesian neural network induce global stationarity in the Gaussian Process induced by an infinite-width limit. Stationary covariance functions in GP can be guaranteed whenever the activation function is chosen to be periodic (beyond sinusoidal). In a concrete example,  periodic activation functions combined with student-t distributed readout parameters result in a Matern class of covariance functions. The paper goes on to show experiments demonstrating that these stationary models are good for capturing uncertainty for out-of-domain data.

The reviewers pointed out the fact that periodic activation functions inducing stationary kernels  may not be mathematically surprising. However, some reviewers also argued that, methodologically, this paper indeed describes a simple solution to an important and challenging problem of ensuring good uncertainty prediction for neural networks.

There were initial concerns regarding the approximation error of ignoring higher frequency effects. However, in the discussion period, authors have clarified the issue 1) providing alternative perspective 2) adding error estimates induced by approximation. Moreover the authors point out that numerical verification in the paper also indicates in reality error is indeed small. In the end, all reviewers agree that the paper should be accepted.

Overall, the reviewers agree that the paper is clearly written with well performed experiments and comprehensive literature review.  The paper would be interesting to various NeurIPS sub-communities e.g. coordinated-based networks (implicit representations), Bayesian deep learning, infinite-width theory.

**Consistency Experiment:**

NeurIPS has a long history of experimentation. In 2014, NeurIPS ran an experiment in which 10% of submissions were reviewed by two independent committees to quantify the randomness in the review process. This year, we repeated a variant of this experiment to see how the quality of the review process has changed over time.  This paper was part of the experiment and was therefore assigned to two committees (consisting of reviewers, an Area Chair, and a Senior Area Chair) that reached independent decisions.  If both committees made the same recommendation, this recommendation was followed. If a single committee recommended acceptance, the paper was accepted (with the exception of a few cases in which the other committee identified what we considered a fatal flaw, e.g., an error in a key result).

Both committees reached the same decision: **Accept (Poster)**

The other committee assigned to the paper recommended **Accept (Poster)**.  You can find the other set of reviews, along with any follow up discussion with the authors here:
https://openreview.net/forum?id=gRwh5HkdaTm